# Do ReLU Networks Have An Edge When Approximating Compactly-Supported Functions?

**Anastasis Kratsios**  *kratsioa@mcmaster.ca*
*Department of Mathematics*
*McMaster University*
*1280 Main Street West, Hamilton, Ontario, L8S 4K1, Canada*

**Behnoosh Zamanlooy**  *zamanlob@mcmaster.ca*
*Department of Computing and Software*
*McMaster University*
*1280 Main Street West, Hamilton, Ontario, L8S 4K1, Canada*

Reviewed on OpenReview: https://openreview.net/forum?id=sNxNi54B8b

## Abstract

We study the problem of approximating compactly-supported integrable functions while implementing their support set using feedforward neural networks. Our first main result transcribes this "structured" approximation problem into a universality problem. We do this by constructing a refinement of the usual topology on the space $L^1_{\text{loc}}(\mathbb{R}^d, \mathbb{R}^D)$ of locally-integrable functions in which compactly-supported functions can only be approximated in $L^1$-norm by functions with matching discretized support. We establish the universality of ReLU feedforward networks with bilinear pooling layers in this refined topology. Consequentially, we find that ReLU feedforward networks with bilinear pooling can approximate compactly supported functions while implementing their discretized support. We derive a quantitative uniform version of our universal approximation theorem on the dense subclass of compactly-supported Lipschitz functions. This quantitative result expresses the depth, width, and the number of bilinear pooling layers required to construct this ReLU network via the target function's regularity, the metric capacity and diameter of its essential support, and the dimensions of the inputs and output spaces. Conversely, we show that polynomial regressors and analytic feedforward networks are not universal in this space.

## 1 Introduction

The variety of available deep learning architectures used in practice and studied in the literature can make it difficult to identify which model is best for a given learning task. In this paper, we consider the problem of approximating an *essentially compactly-supported (Lebesgue) integrable function* $f : \mathbb{R}^d \to \mathbb{R}^D$ using a the rudimentary feedforward architecture. The typical example of such a map is the *distance function* to the complement of compact subset of $K \subseteq \mathbb{R}^d$, defined by

$$x \mapsto d_{\mathbb{R}^d \setminus K}(x) := \inf_{z \in \mathbb{R}^d \setminus K} \|z - x\|;$$

where $K$ is non-empty. These maps are common in computer vision Di Gesu & Starovoitov (1999), in computational physics Tsai (2002), and they are used for partitioning latent metric subspaces of $\mathbb{R}^d$ (see Cobzaş et al. (2019)).

Since an essentially compactly-supported integrable function contains more structure than an arbitrary locally-integrable function; namely, its essential support set, it is natural to ask if we can *approximate such a function to arbitrary precision* while simultaneously *exactly implementing its support up to a discretization of the input space* $\mathbb{R}^d$. Even if we only focus on the class of feedforward networks from $\mathbb{R}^d$ to $\mathbb{R}^D$ it can be unclear which activation function produces feedforward networks which are compatible with this objective.

We can immediately rule-out networks built using any combined number of analytic activation functions, by virtue of their analyticity. Examples include the *sigmoid* activation function, the Swish activation function of Ramachandran et al. (2018), the GeLU activation of Hendrycks & Gimpel (2016), the Softplus non-linearity Glorot et al. (2011), the sin function used in SIREN networks Sitzmann et al. (2020), tanh, Hermite polynomial activation functions used in Ma & Khorasani (2005), and several others examples. Since, the composition of analytic functions is again an analytic function then every such neural network must be analytic. The trouble is that no analytic function can simultaneously be compactly-supported and non-zero. Therefore, no feedforward architecture using only analytic activation functions can approximate a compactly-supported function in while exactly implementing its support (up to a discretization of the input space $\mathbb{R}^d$).

We therefore turn our attention to other most common class of activation functions; namely, (non-affine) piecewise linear activation functions such as the ReLU nonlinearity of Fukushima (1969), the PReLU activation function of He et al. (2015), or the leaky ReLU function of Maas et al. (2013). Since this class of activation functions is not analytic, it is at-least possible for neural networks with piecewise linear class to approximate essentially-compactly supported integrable functions in the aforementioned sense. Since every neural network with a piecewise linear activation function can be implemented by a deep ReLU network (see (Yarotsky, 2017b, Proposition 1)) and since the ReLU activation function, defined by $\mathrm{ReLU}(x) \overset{\text{def.}}{=} \max\{0, x\}$, vanishes on a large part of its input space then, it is plausible that such neural networks can approximate a function while themselves having compact support. Thus, feedforward neural neural networks with (non-affine) piecewise linear activation functions seem to be a viable candidate for solving this approximation-theoretic problem.

In this paper, we demonstrate that deep feedforward networks with (non-affine) piecewise linear activation functions can approximate any essentially compactly-supported (Lebesgue) integrable function $f : \mathbb{R}^d \to \mathbb{R}^D$ while simultaneously exactly implementing its support up to a discretization of $\mathbb{R}^d$, provided that the feedforward model can also leverage bilinear pooling layers. We denote this set of functions by $\mathrm{NN}^{\mathrm{ReLU+Pool}}$.

To answer this question we construct a topology $\tau$ on the set of locally-integrable functions $L^1_{\mathrm{loc}}(\mathbb{R}^d, \mathbb{R}^D)$ from $\mathbb{R}^d$ to $\mathbb{R}^D$ formalizing the mode of approximation for essentially-compactly-supported integrable functions outlined thus far. Furthermore, we wish that our universal approximation theorem implies the classical notion of $L^1$-universal approximation derived in (Hornik et al., 1989; Yarotsky, 2018; Gühring et al., 2020; Lu et al., 2021; Shen et al., 2022; Opschoor et al., 2022); therefore, our topology is constructed as a refinement of the usual metric topology on $L^1_{\mathrm{loc}}(\mathbb{R}^d, \mathbb{R}^D)$ as well as the familiar norm topology on the subset $L^1(\mathbb{R}^d, \mathbb{R}^D)$ of (globally) Lebesgue-integrable functions. Our first main result confirms that $\tau$ is well-defined and that it encodes the aforementioned behaviour of models approximating compactly supported functions.

**Theorem 1** (Approximation of Essentially Compactly-Supported Lebesgue-Integrable Functions in $\tau$)**.**
*There is a strict refinement $\tau$ of the topology on $L^1_{\mathrm{loc}}(\mathbb{R}^d, \mathbb{R}^D)$ which refines the metric topology on $L^1_{\mathrm{loc}}(\mathbb{R}^d, \mathbb{R}^D)$, whose restriction to $L^1(\mathbb{R}^d, \mathbb{R}^D)$ is also a strict refinement of the $L^1$-norm topology, and satisfies:*

*(i)* ***Approximation of Compactly Supported Functions is Only Possible with Compactly Supported Models:***
*For every $n \in \mathbb{N}_+$ and every $f \in L^1(\mathbb{R}^d, \mathbb{R}^D)$ which is essentially supported on $[-n, n]^d$, a sequence $\{f_k\}_{k \in \mathbb{N}^+}$ in $L^1_{\mathrm{loc}}(\mathbb{R}^d, \mathbb{R}^D)$ converges to $f$ with respect to $\tau$ only if there is an $N \in \mathbb{N}_+$ with $N \geq n$ such that all but a finite number of $f_k$ are in $[-N, N]^d$ and $\lim_{k \uparrow \infty} \|f_k - f\| = 0$.*

*(ii)* ***Simultaneous Discretized Support Implementation and $L^1$-Approximation Imply $\tau$-universality:*** *A subset $\mathscr{F}$ of $L^1_{\mathrm{loc}}(\mathbb{R}^d, \mathbb{R}^D)$ is dense for $\tau$ if, for every $f \in \mathrm{Lip}_{\mathrm{loc}}(\mathbb{R}^d, \mathbb{R}^D)$ which is essentially compactly supported, there is a sequence $\{f_n\}_{n=1}^\infty$ in $\mathscr{F}$ satisfying*

$$\lim_{n \uparrow \infty} \|f_n - f\|_{L^1(\mathbb{R}^d, \mathbb{R}^D)} = 0 \text{ and } \mathrm{ess\text{-}supp}(f) \cup \bigcup_{n=1}^\infty \mathrm{ess\text{-}supp}(f_n) \subseteq [-n_f - 1, n_f + 1]^d;$$

*where $n_f \overset{\text{def.}}{=} \min\{n \in \mathbb{N}_+ : \mathrm{ess\text{-}supp}(f) \subseteq [-n, n]^d\}$.*

*(iii)* ***Non Implementability Restrictions:*** *For every $f \in \mathrm{NN}^{\sigma_{\mathrm{PW\text{-}Lin}}+\mathrm{Pool}}$ the set $\{f\}$ it not open in $\tau$.*

We call the topology $\tau$ constructed in the proof of Theorem 1 the compactly-supported $L^1$-topology (cs$L^1$-topology).

**The Qualitative Effect Encoded by the CSL$^1$-Topology $\tau$**

Convergence to a compactly supported Lipschitz function (such as $f$) in the csL$^1$-topology $\tau$ requires simultaneous approximation of $f$'s value and correct implementation of its support, instead of only requiring that $f$'s values are approximated as in the topologies on $L^1(\mathbb{R}^d, \mathbb{R}^D)$ and on $L^1_{\text{loc}}(\mathbb{R}^d, \mathbb{R}^D)$.

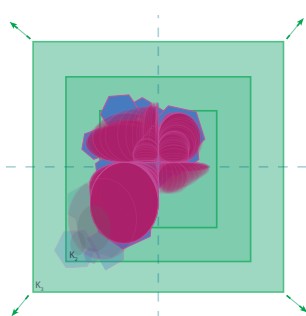

Figure 1: Approximation of a compactly supported Lipschitz function by a ReLU network with bilinear pooling

The two-dimensional example is illustrated by Figure 1, which shows the target function $f: \mathbb{R}^d \overset{\text{def.}}{=} \mathbb{R}^2 \to \mathbb{R}$ (illustrated in red), an approximation of it by a ReLU network $\hat{f}$ with bilinear pooling (illustrated in blue), and a discretization given by a suitable of compact subsets $\{K_n\}_{n=1}^{\infty}$ of $\mathbb{R}^d$ covering $\mathbb{R}^d$ up to a set of Lebesgue measure 0. The target function's value and the network's output are represented by the vividness (alpha) of the each respective color. We see that ReLU network $\hat{f}$ with bilinear pooling is simultaneously close to the target function $f$'s value and that $\hat{f}$ identifies the correct number of compacts subsets $\{K_1, K_2, K_3\}$ containing target function $f$ is supported (possibly with one extra set; in this case $K_3$). Moreover, somewhat surprisingly, we will see that this approximation guarantee is independently of our discretization of $\mathbb{R}^d$ (i.e. our choice of suitable compact subsets $\{K_n\}_{n=1}^{\infty}$ of $\mathbb{R}^d$).

This illustration is formalized by the following strengthened universal approximation theorem which shows that $\text{NN}^{\text{ReLU+Pool}}$ is universal in the topology $\tau$ on $L^1_{\text{loc}}(\mathbb{R}^d, \mathbb{R}^D)$. Rigorously, we call a function $\sigma \in C(\mathbb{R})$ is said to be non-affine and piecewise linear if $\mathbb{R}$ can be covered by a sequence of intervals on which $\sigma$ is affine and there is at-least one point at which $\sigma$ is not differentiable. Let $\sigma_{\text{PW-Lin}}$ be a non-affine piecewise linear activation function and let $\text{NN}^{\text{ReLU+Pool}}$ denote the set of deep feedforward networks mapping $\mathbb{R}^d$ to $\mathbb{R}^D$ with bilinear pooling layer, defined by $\text{Pool}(x_1, \ldots, x_{2n}) \overset{\text{def.}}{=} (x_1 x_2, \ldots, x_{2n-1} x_{2n})$, at their output.

**Theorem 2** (Universal Approximation Theorem + Support Implementation for Compactly Supported Functions).
*Let $\log_2(d) \in \mathbb{N}_+$ and let $\tau$ be the topology on $L^1_{\text{loc}}(\mathbb{R}^d, \mathbb{R}^D)$ from Theorem 1. If $\sigma_{\text{PW-Lin}} \in C(\mathbb{R})$ is piecewise linear with at-least 2 pieces then $\text{NN}^{\sigma_{\text{PW-Lin}}+\text{Pool}}$ is dense in $L^1_{\text{loc}}(\mathbb{R}^d, \mathbb{R}^D)$ with respect to $\tau$.*

This *qualitative* universal approximation theorem confirms that (non-affine) piecewise linear neural networks can approximate essentially-compactly supported Lebesgue integrable functions between Euclidean spaces while exactly implementing their discretized support. However, the result does not describe the complexity of the neural networks model. Therefore, we also derive a quantitative version of Theorem 2 specialized for the dense class of compactly-supported Lipschitz functions mapping $\mathbb{R}^d$ to $\mathbb{R}^D$; where density is meant with respect to the topology $\tau$.

**Quantitative Approximation in the CSL$^1$-Topology $\tau$**

Once $\tau$ is constructed, the crux of our analysis when proving Theorem 2 reduces to obtaining a *quantitative* "structured" universal approximation result shows that given any compactly supported Lipschitz function $f: \mathbb{R}^d \to \mathbb{R}^D$ we identify a neural network $\hat{f} \in \text{NN}^{\text{ReLU+Pool}}$ which can approximate $f$'s value while also implementing its discretized support.

A rigorous statement of our result requires some terminology. Denote the $d$-dimensional Lebesgue measure by $\mu$. The *essential support* of a $f \in L^1_{\text{loc}}(\mathbb{R}^d, \mathbb{R}^D)$ is defined by $\text{ess-supp}(f) \overset{\text{def.}}{=} \mathbb{R}^d - \bigcup \{U \subseteq \mathbb{R}^d : U \text{ open and } \|f\|(x) = 0 \ \mu\text{-a.e. } x \in U\}$. We say that an $f \in L^1_{\text{loc}}(\mathbb{R}^d, \mathbb{R}^D)$ is *essentially compactly supported* if $\text{ess-supp}(f)$ is contained in a closed and bounded subset of $\mathbb{R}^d$. The regularity of a Lipschitz

function $f : \mathbb{R}^d \to \mathbb{R}^D$ (i.e. a function with at-most linear growth) is quantified by its Lipschitz constant $\text{Lip}(f) \stackrel{\text{def.}}{=} \sup_{x_1, x_2 \in \mathbb{R}^d, x_1 \neq x_2} \frac{\|f(x_1)-f(x_2)\|}{\|x_1 - x_2\|}$. The "complexity" of a subset $X \subseteq \mathbb{R}^d$ is quantified both in terms of its size $\text{diam}(X) \stackrel{\text{def.}}{=} \sup_{x_1, x_2 \in X} \|x_1 - x_2\|$ and its "fractal dimension" as quantified by its *metric capacity* defined by

$$\text{cap}(X) \stackrel{\text{def.}}{=} \sup \left\{ n \in \mathbb{N}_+ : (\exists x_1, \ldots, x_n \in X), (\exists r > 0) \sqcup_{i=1}^N B_2(x_i, r/5) \subset B_2(x_0, r) \right\},$$

where $\sqcup$ denotes the union of *disjoint* subsets of $\mathbb{R}^d$ and where $B_2(x, r) \stackrel{\text{def.}}{=} \{u \in \mathbb{R}^d : \|u - x\| < r\}$. We mention that, for a compact Riemannian manifold, the $\log_2$-metric capacity is always a multiple of the manifold's topological dimension and the $\log_2$-metric capacity of a $d$-dimensional cube in $\mathbb{R}^d$ is proportional to $d$; (see (Acciaio et al., 2022, 2.1.3) for further details). We denote the set of polynomial functions from $\mathbb{R}^d$ to $\mathbb{R}^D$ by $\mathbb{R}[x_1, \ldots, x_d : D]$.

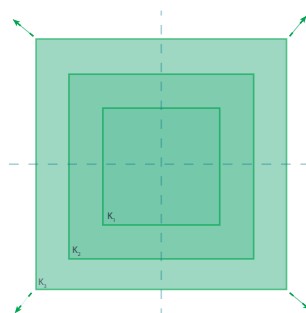

Figure 2: The Cubic-Annuli Discretization of $\mathbb{R}^d$ (Definition 1)

Unlike classical quantitative universal approximation theorems, this next result describes the width, depth, and number of bi-linear pooling layers required for a neural network in $\text{NN}^{\text{ReLU+Pool}}$ to approximate a compactly supported Lipschitz functions while simultaneously exactly implementing its support; up to the following standardized discretization of the input space $\left\{ K_n \stackrel{\text{def.}}{=} \{x \in \mathbb{R}^d : n < \|x\|_\infty \leq n+1\} \right\}_{n=1}^\infty$, illustrated in Figure 2.

Let us mention that, using a using category-theoretic argument, we show that the $\text{csL}^1$-topology $\tau$ is independent of any discretization of $\mathbb{R}^d$ used to construct it. Consequentially, all our universality arguments and statements, such as Theorem 2, can without loss of generality be formulated using the standardized discretization illustrated in Figure 2.

**Theorem 3** (Support Implementation and Uniform + $\tau$ Approximation of ReLU Networks with Pooling)**.**
*Let $f : \mathbb{R}^d \to \mathbb{R}^D$ be Lipschitz and compactly-supported and $\log_2(d) \in \mathbb{N}_+$. For every "width parameter" $N \in \mathbb{N}_+$ and every sequence $\{\varepsilon_n\}_{n=1}^\infty$ in $(0, \infty)$ converging to $0$, there is a sequence $\{\hat{f}^{(n)}\}_{n=1}^\infty$ in $\text{NN}^{\text{ReLU+Pool}}$ satisfying:*

- (i) ***Quantitative Worst-Case Approximation:*** *for each $n \in \mathbb{N}_+$ $\max_{x \in [n_f, n_f]^d} \left\| f(x) - \hat{f}^{(n)}(x) \right\| \leq \varepsilon_n$,*

- (ii) ***Convergence in $\text{CSL}^1$-Topology $\tau$:*** *$\{\text{Pool} \circ \hat{f}^{(n)}\}_{n=1}^\infty$ converges to $f$ in the $\text{csL}^1$-topology $\tau$,*

- (iii) ***Support Implementation:*** *$\text{ess-supp}(\hat{f}^{(n)}) \subseteq \left[ -\sqrt[d]{2^{-d}\varepsilon_n + n_f^d}, \sqrt[d]{2^{-d}\varepsilon_n + n_f^d} \right]^d$, where $n_f$ is defined by $n_f \stackrel{\text{def.}}{=} \min\{n \in \mathbb{N}_+ : \text{ess-supp}(f) \subseteq [-n, n]^d\}$.*

*Moreover, each $\hat{f}^{(n)}$ is specified by:*

- (iv) ***Width:*** *$\hat{f}^{(n)}$ has Width $C_3 + C_4 \max\{d\lfloor N^{1/d} \rfloor, N + 1\}$,*

- (v) ***Depth:*** *$\hat{f}^{(n)}$ has Depth $\frac{\varepsilon_n^{-d/2}}{N \log_3(N+2)^{1/2}} \left( \log_2(\text{cap}(\text{ess-supp}(f))) \text{diam}(\text{ess-supp}(f)) \text{Lip}(f) \right)^d C_1 + C_2$*

- (vi) ***Number of Bilinear Pooling Layers:*** *$\hat{f}^{(n)}$ uses $\log_2(d) + 1$ bilinear pooling layers.*

where the dimensional constants are $C_1 \stackrel{\text{def.}}{=} c\, 2^d D^{3/d} d^d + 3d$, $C_2 \stackrel{\text{def.}}{=} +2d+2$, $C_3 \stackrel{\text{def.}}{=} \max\{d(d-1)+2, D\}$, $C_4 \stackrel{\text{def.}}{=} d(D+1)+3^{d+3}$, and where $c > 0$ is an absolute constant independent of $X, d, D$, and $f$.

In addition to the main contribution of Theorem 3, there are several additional points of technical novelty in Theorem 3. The first such point is that the network complexity depends on the *metric capacity* of the target function's essential support. Omitting constants, the depth of the ReLU networks $\hat{f}^{(n)}$ with pooling in Theorem 3 encodes three of the target function $f$'s *structural attributes*. The first is the desired approximation quality, with more depth translating to better approximation capacity, and the second is the target function's regularity; both these factors are present in most available quantitative approximation theorems (Yarotsky, 2017b; Gühring et al., 2020; Jiao et al., 2021; Lu et al., 2021; Shen et al., 2022; Opschoor et al., 2022).

$$
\text{Depth}(f^{(n)}) \approx \underbrace{\frac{\varepsilon_n^{-d/2}}{N \log_3(N+2)^{1/2}}}_{\text{Approximation Quality}} \quad \underbrace{\text{Lip}(f)^d}_{\text{Target's Regularity}} \quad \underbrace{\Big(\log_2(\text{cap}(\text{ess-supp}(f)))\,\text{diam}(\text{ess-supp}(f))\Big)^d}_{\text{Complexity: Target's Essential Support}}
$$

Part of the novelty of Theorem 3 is that it identifies a third quantity impacting the approximation quality of a ReLU network with pooling; namely, the complexity of the target function's support. This third factor can be decomposed into two parts, the diameter of the target function's essential support, which other approximation theorems have also considered Siegel & Xu (2020); Kratsios & Papon (2022), but what is most interesting here is the effect of the *fractal dimension* (via the metric capacity; see Bruè et al. (2021) for details) of the target function's essential support. In particular, the result shows that functions essentially supported on low-dimensional sets (e.g. low-dimensional latent manifolds) must be simpler to approximate than those with unbounded support (e.g. locally Lebesgue-integrable functions supported on $\mathbb{R}^d$. Theorem 2 and variant of (Shen et al., 2022, Theorem 1.1) and of the main result of Yarotsky (2017b) where the approximation has *controlled support* made to match that of the target function. To the best of the authors' knowledge, the result is also the only quantitative universal approximation which encodes the target function $f$'s complexity in terms of its Lipschitz regularity, as well as, the size and dimension of its essential support. We note that, since one can show that $\tau$ is not a metric topology, therefore, a *quantitative* counterpart of Theorem 2 does not exist for arbitrary $f \in L^1_{\text{loc}}(\mathbb{R}^d, \mathbb{R}^D)$ for the csL$^1$-topology $\tau$.

An additional point of technical novelty in Theorem 3 appears through new tools to the deep learning literature used in the result's derivation. Namely, we introduce the non-affine random projections of Ohta (2009); Bruè et al. (2021) to encode this dependence into the approximation using contemporary Lipschitz-extension arguments when deriving the universal approximation theorem. We note that, these random projections are distinct mathematical objects from the linear random projections of Johnson & Lindenstrauss (1984) and, as shown in Ambrosio & Puglisi (2020), these random projections are closely related to the random partitions of unity introduced by Lee & Naor (2005).

**A Sanity Check: Comparison Between Networks in $\text{NN}^{\text{ReLU+Pool}}$ and Analytic Models**
We round off our discussion by verifying our intuition about analytic model classes is indeed reflected by the topology csL$^1$-topology $\tau$. For illustrative purposes, we first consider the classical polynomial regressors, whose universal approximation capabilities in more classical topologies are guaranteed by the classical Stone-Weierstrass theorem and its numerous contemporary variants Prolla (1994), Timofte et al. (2018), or of Galindo & Sanchis (2004)).

**Proposition 1** (Polynomial-Regressors Are Not Universal $L^1_{\text{loc}}(\mathbb{R}^d, \mathbb{R}^D)$ for $\tau$)**.**
*The set $\mathbb{R}[x_1, \ldots, x_d : D]$ is not dense in $L^1_{\text{loc}}(\mathbb{R}^d, \mathbb{R}^D)$ for the CSL$^1$-topology $\tau$.*

We return to our motivational example, by confirming that the class of deep feedforward networks which can leverage any number of analytic functions and which also have access to bilinear pooling are not dense in $L^1_{\text{loc}}(\mathbb{R}^d, \mathbb{R}^D)$ for $\tau$. Denote this class of neural networks by $\text{NN}^{\omega+\text{Pool}}$; which reflects the notation for the set $C^\omega(\mathbb{R})$ of real-valued analytic functions on $\mathbb{R}$.

**Proposition 2** (Analytic Feedforward Networks Are Not Universal $L^1_{\text{loc}}(\mathbb{R}^d, \mathbb{R}^D)$ for $\tau$)**.**
*The set $\text{NN}^{\omega+\text{Pool}}$ is not dense in $L^1_{\text{loc}}(\mathbb{R}^d, \mathbb{R}^D)$ with respect to the CSL$^1$-topology $\tau$ $\tau$.*

## 1.1 Connection to Other Deep Learning Literature

Our results are perhaps most closely related to Park et al. (2021) which demonstrates, to the best of our knowledge, the only other qualitative gap in the deep learning theory. Namely, therein, the authors identify a minimum width under which all networks become too narrow to approximate any integrable function; equivalently, the set of "very narrow" deep feedforward networks is qualitatively less expressive than the set of "arbitrary deep feedforward networks". Just as our main results are qualitative, the results of Park et al. (2021) can be contrasted against the main result of Shen et al. (2022) which quantifies the exact impacts of depth and width on approximation error of deep feedforward networks.

Our results also add to the recent scrutiny given to deep feedforward networks deploying several activation functions (Jiao et al., 2021; Yarotsky & Zhevnerchuk, 2020; Beknazaryan, 2021; Yarotsky, 2021; Acciaio et al., 2022). The connection to this branch of deep learning theory happens on two distinct fronts. First $NN^{\omega+Pool}$ is clearly a family of deep feedforward networks simultaneously utilizing several activation functions. However, more interesting, is the second connection between networks in $NN^{ReLU+Pool}$ and the approximation theory of deep feedforward networks with generalized ReLU activation function $ReLU_r(x) \stackrel{\text{def.}}{=} \max\{x,0\}^r$, where $r \in \mathbb{R}$ is a *trainable parameter*. This is because, Pool can be implemented by a feedforward network with $ReLU_2$ activation function, since $x^2 = ReLU_2(x) + ReLU_2(-x)$ (where $x \in \mathbb{R}$) and (Kidger & Lyons, 2020, Lemma 4.3) shows that the multiplication map $\mathbb{R}^2 \ni (x_1, x_2) \mapsto x_1 x_2$ can be exactly implemented by a neural network with one hidden layer and with activation function $x \mapsto x^2$. Therefore, any $f \in NN^{ReLU+Pool}$ there are $f_1, \ldots, f_I \in NN^{ReLU_2} \cup NN^{ReLU}$ representing $f$ via

$$f = f_I \circ \cdots \circ f_1.$$

We note that networks with activation function in $\{ReLU_r\}_{r \in \mathbb{R}}$ have recently rigorous study in Gribonval Rémi et al. (2021) and are related to the the constructive approximation theory of splines where $ReLU_r$ are known as *truncated powers* (see (DeVore & Lorentz, 1993, Chapter 5, Equation (1.1))). We also mention that Theorem 3 is related to recent deep learning research considering the approximation of a function or probability measure's support. The former case is considered by Kratsios & Zamanlooy (2022), where the authors consider an exotic neural network architecture specialized in the approximation of piecewise continuous functions in a certain sense. In the latter case, Puthawala et al. (2022) use a GAN-like architecture to approximate probability distributions supported on a low-dimensional manifold by approximating their manifold and the density thereon using a specific neural network architecture. In contrast, our results compare the approximation capabilities of feedforward networks built using different activation functions.

### Organization of Paper

This paper is organized as follows. Section 2 reviews the necessary deep learning terminology, measure theoretic, and topological background needed in the formulation of our main result. Section 4 derives the main results, with the understanding that all technical lemmata and their proofs are relegated to the paper's appendix. Section 5 then discusses some of the implications our results and possible future directions of this type of analysis.

## 2 Preliminaries

We use $\mathbb{N}_+$ to denote the set of positive integers, fix $d, D \in \mathbb{N}_+$, and let $\|\cdot\|$ denote Euclidean distance on $\mathbb{R}^D$.

To simplify the analysis, we emphasize that $d$ will *always be assumed to be a power of* 2; *i.e.* $d = 2^{d'}$ *where* $d' \in \mathbb{N}_+$.

### 2.1 Deep Feedforward Networks

Originally introduced by McCulloch & Pitts (1943) as a prototypical model for artificial neural computation, deep feedforward networks have since lead to computational breakthroughs across various areas from biomedical imaging Ronneberger et al. (2015) to quantitative finance Buehler et al. (2019); Jaimungal (2022). Though deep learning tools has become pedestrian in most contemporary scientific computational endeavors, the mathematical foundations of deep learning are still in their early stages.

Therefore, in this paper, we study the approximation-theoretic properties of what is arguably the most basic deep learning model; namely, the *feedforward (neural) network*. These are models which iteratively process inputs in $\mathbb{R}^d$ by repeatedly applying affine transformations (as in linear regression) and simple component-wise non-linearity called *activation functions*, until an output in $\mathbb{R}^D$ is eventually produced.

Our discussion naturally begins with the formal definition of the class of deep feedforward neural networks defined by a (non-empty) *family of (continuous) activation functions* $\Sigma \subseteq C(\mathbb{R})$. In the case where $\Sigma = \{\sigma\}$ is a singleton, one recovers the classical definition of a feedforward network studied in Cybenko (1989); Hornik et al. (1989); Leshno et al. (1993); Yarotsky (2017b); Kidger & Lyons (2020) and when $\Sigma = \{\sigma_r\}_{r \in \mathbb{R}}$ and the map $(r,x) \mapsto \sigma_r(x)$ is Lebesgue a.e. differentiable then one obtains so-called *trainable activation functions* as considered in Cheridito et al. (2021a); Kratsios et al. (2022); Acciaio et al. (2022) of which the $\mathrm{PReLU}_r(x) \stackrel{\text{def.}}{=} \max\{x, rx\}$ activation function of He et al. (2015) is prototypical. More broadly, neural networks build using families of activation functions $\Sigma$ exhibiting sub-exponential approximation rates have also recently become increasingly well-studied; e.g. Yarotsky & Zhevnerchuk (2020); Jiao et al. (2021); Yarotsky (2021); Beknazaryan (2021).

Consider the *bilinear pooling layer*, from computer vision (Lin et al., 2015; Kim et al., 2016; Fang et al., 2019), given for any *even* $n \in \mathbb{N}_+$ and $x \in \mathbb{R}^n$ as

$$\mathrm{Pool}(x) \stackrel{\text{def.}}{=} \left(x_i x_{n/2+i}\right)_{i=1}^{n/2}.$$

Alternatively, Pool can be thought of as a *masking layer* with non-binary values, similar to the bilinear masking layers or bilinear attention layers used in the computer-vision literature Fang et al. (2019); Lin et al. (2015) or in the low-rank learning literature Kim et al. (2016), or as the *Hadamard product* of the first $n/2$ components of a vector in $\mathbb{R}^n$ with the last $n$ components.

Fix a *depth* $J, d, D \in \mathbb{N}_+$. A function $\hat{f}: \mathbb{R}^d \to \mathbb{R}^D$ is said to be a *deep feedforward network with (bilinear) pooling* if for every $j = 0, \ldots, J-1$ there are Boolean *pooling parameters* $\alpha^{(j)} \in \{0,1\}$, $d_{j,2} \times d_{j,1}$-dimensional matrices $A^{(j)}$ with $d_{j+1,1}/2 = d_{j,2}$ if $d_{j,2}$ is even and if $\alpha = 1$ and $d_{j+1,1} = d_{j,2}$ otherwise which are called *weights*, $b^{(j)} \in \mathbb{R}^{d_j}$ and a $c \in \mathbb{R}^{d_J}$ called *biases*, and *activation functions* $\sigma^{(j,i)} \in \Sigma$ such that $\hat{f}$ admits the iterative representation

$$
\begin{aligned}
\hat{f}(x) &\stackrel{\text{def.}}{=} x^{(J)} + c \\
x^{(j+1)} &\stackrel{\text{def.}}{=} \begin{cases} \mathrm{Pool}(\tilde{x}^{(j+1)}) & : \alpha^{(j)} = 1 \text{ and } d_{j+1} \text{ is even} \\ \tilde{x}^{(j+1)} & : \text{else} \end{cases} \qquad \text{for } j = 0, \ldots, J-1 \\
\tilde{x}_i^{(j+1)} &\stackrel{\text{def.}}{=} \sigma^{(j,i)}\big((A^{(j)} x^{(j)} + b^{(j)})_i\big) \qquad \text{for } j = 0, \ldots, J-1; \, i = 1, \ldots, d_{j+1} \\
x^{(0)} &\stackrel{\text{def.}}{=} x.
\end{aligned}
\tag{1}
$$

We denote by $\mathrm{NN}^{\Sigma+\mathrm{Pool}}$ the set of all deep feedforward networks with pooling and activation functions belonging to $\Sigma$. If, in the above notation, $\hat{f}$ such that $x^{(j+1)} = \tilde{x}^{(j+1)}$ then, we say that $\hat{f}$ is a *deep feedforward network (without pooling)*. The collection of all deep feedforward networks (without pooling) is denoted by $\mathrm{NN}^\Sigma$ and activation functions belonging to $\Sigma$.

In either case, if $\Sigma$ consists only of a single activation function $\sigma$ then, we use $\mathrm{NN}^{\Sigma+\mathrm{Pool}}$ to denote $\mathrm{NN}^{\sigma+\mathrm{Pool}}$. Similarly, if $\Sigma = \{\sigma\}$ then we set $\mathrm{NN}^\sigma \stackrel{\text{def.}}{=} \mathrm{NN}^\Sigma$. Let us consider some examples of activation functions.

**Example 1** (Non-Affine and Piecewise Linear Networks). *An activation function $\sigma \in C(\mathbb{R})$ is called non-affine and piecewise linear if: there exist $-\infty = t_0 < t_1 < \cdots < t_p < t_{p+1} = \infty$ and some $m_1, \ldots, m_p, b_1, \ldots, b_p \in \mathbb{R}$ for which*

   *(i) $\sigma(x) = m_i x + b_i$ for every $t \in (t_i, t_{i+1})$ for each $i = 0, \ldots, p$,*

   *(ii) There exist some $i \in \{1, \ldots, p\}$ for which $\sigma'(t_i)$ is undefined.*

*The prototypical example of such an activation function is $\mathrm{ReLU}(x) \stackrel{\text{def.}}{=} \max\{0, x\}$.*

**Example 2** (Deep Feedforward Networks with "Adaptive" Analytic Activation Functions ($\mathrm{NN}^\omega$)). *Let $C^\omega(\mathbb{R})$ denote the set of a analytic maps from $\mathbb{R}$ to itself. We set $\mathrm{NN}^\omega \stackrel{\text{def.}}{=} \mathrm{NN}^{C^\omega(\mathbb{R})}$ and we use $\mathrm{NN}^{\omega+\mathrm{Pool}} \stackrel{\text{def.}}{=} \mathrm{NN}^{C^\omega(\mathbb{R})+\mathrm{Pool}}$*

## 2.2 Measure Theory

Following (Schwartz, 1966, Chapter 1), we call Borel measurable function $f : \mathbb{R}^d \to \mathbb{R}^D$ is called *locally integrable* if, on each compact subset $K \subset \mathbb{R}^d$ the Lebesgue integral $\int_{x \in K} \|f(x)\| \, dx$ is finite. Let $L^1_{\text{loc}}(\mathbb{R}^d, \mathbb{R}^D)$ denote the set of locally integrable functions from $\mathbb{R}^d$ to $\mathbb{R}^D$; with equivalence relation $f \sim g$ if and only if $f$ and $g$ differ only on a set of Lebesgue measure 0. The set $L^1_{\text{loc}}(\mathbb{R}^d, \mathbb{R}^D)$ is made into a *complete metric space* by equipping it with the distance function $d_{L^1_{\text{loc}}}$ defined on any two $f, g \in L^1_{\text{loc}}(\mathbb{R}^d, \mathbb{R}^D)$ by

$$d_{L^1_{loc}}(f,g) \stackrel{\text{def.}}{=} \sum_{n=1}^{\infty} \frac{1}{2^n} \frac{\int_{\|x\| \leq n} \|(f(x) - g(x))\| \, dx}{1 + \int_{\|x\| \leq n} \|(f(x) - g(x))\| \, dx}.$$

The subset of $L^1_{\text{loc}}(\mathbb{R}^d, \mathbb{R}^D)$ consisting of all *integrable "functions"*, i.e. all $f \in L^1_{\text{loc}}(\mathbb{R}^d, \mathbb{R}^D)$ for which the integral $\int_{x \in \mathbb{R}^d} \|f(x)\| \, dx$ is finite, is denoted by $L^1(\mathbb{R}^d, \mathbb{R}^D)$. The set $L^1(\mathbb{R}^d, \mathbb{R}^D)$ is made into a Banach space, called the *Bochner-Lebesgue* space, by equipping it with the norm $\|f\|_{L^1} \stackrel{\text{def.}}{=} \int_{x \in \mathbb{R}^d} \|f(x)\| \, dx$.

## 2.3 Point-Set Topology

In most of analysis one uses the language of *metric spaces*, i.e.: an (abstract) set of points $X$ together with a distance function $d : X^2 \to [0, \infty)$ satisfying certain axioms (see (Heinonen, 2001)), to the similarity of dissimilarity between different mathematical objects. However, not all notions of similarity can be described by a metric structure and this is in particular true for several very finer notions of similarity playing central roles in functional analysis (see Narayanaswami & Saxon (1986)).

In such situations, one instead turns to the notion of a *topology* to qualify closeness of two objects without relying on the quantitative notion of distance defined though by a metric. Briefly, a topology $\tau_X$ on a set $X$ is a collection of subsets of $X$ declared as being "open"; we require only that $\tau_X$ satisfy certain axioms reminiscent of the familiar open neighborhoods build using balls in metric space theory. Namely, $\tau_X$ contains the empty set and the "total" set $X$, the union of elements in $\tau_X$ are again a member of $\tau_X$, and the countable intersection of sets in $\tau_X$ are again a set in $\tau$. A *topological space* is a pair $(X, \tau_X)$ of a set $X$ and a topology $\tau_X$ on $X$. If clear from the context, we denote $(X, \tau_X)$ by $X$.

**Example 3** (Metric Topology on $L^1_{\text{loc}}(\mathbb{R}^d, \mathbb{R}^D)$). *The metric topology on $L^1_{\text{loc}}(\mathbb{R}^d, \mathbb{R}^D)$, which exists, is the smallest topology on $L^1_{\text{loc}}(\mathbb{R}^d, \mathbb{R}^D)$ containing all the open balls*

$$B_{L^1_{\text{loc}}(\mathbb{R}^d, \mathbb{R}^D)}(f, \varepsilon) \stackrel{\text{def.}}{=} \left\{ g \in L^1_{\text{loc}}(\mathbb{R}^d, \mathbb{R}^D) : d_{L^1_{\text{loc}}}(f, g) < \varepsilon \right\},$$

*where $f \in L^1_{\text{loc}}(\mathbb{R}^d, \mathbb{R}^D)$ and $\varepsilon > 0$. We denote this topology by $\tau_{\text{loc}}$.*

A topology on the subset $L^1(\mathbb{R}^d, \mathbb{R}^D)$ of $L^1_{\text{loc}}(\mathbb{R}^d, \mathbb{R}^D)$ can always be defined by restricting $\tau_{\text{loc}}$ as follows.

**Example 4** (Subspace Topology on $L^1(\mathbb{R}^d, \mathbb{R}^D)$). *The subspace topology on $L^1(\mathbb{R}^d, \mathbb{R}^D)$, relative to the metric topology on $L^1_{\text{loc}}(\mathbb{R}^d, \mathbb{R}^D)$, is the collection $\{U \cap L^1(\mathbb{R}^d, \mathbb{R}^D) : U \in \tau_{\text{loc}}\}$.*

A topology $\tau'_X$ on $X$ is said to be strictly *stronger* than another topology $\tau_X$ on $X$ if $\tau_X \subset \tau'_X$. The key relation between $L^1_{\text{loc}}(\mathbb{R}^d, \mathbb{R}^D)$ and $L^1(\mathbb{R}^d, \mathbb{R}^D)$ is that even if former is strictly larger as a set, the topology on the latter induced by the norm $\|\cdot\|_{L^1}$ is strictly stronger than $\tau_{\text{loc}}$.

The norm topology on $L^1(\mathbb{R}^d, \mathbb{R}^D)$ is defined as follows.

**Example 5** (Norm Topology on $L^1(\mathbb{R}^d, \mathbb{R}^D)$). *The norm topology on $L^1(\mathbb{R}^d, \mathbb{R}^D)$, which exists, is the smallest topology on $L^1(\mathbb{R}^d, \mathbb{R}^D)$ which contains all the open balls*

$$B_{L^1(\mathbb{R}^d, \mathbb{R}^D)}(f, \varepsilon) \stackrel{\text{def.}}{=} \left\{ g \in L^1(\mathbb{R}^d, \mathbb{R}^D) : \|f - g\|_{L^1} < \varepsilon \right\},$$

*where $f \in L^1(\mathbb{R}^d, \mathbb{R}^D)$ and $\varepsilon > 0$. We denote this topology by $\tau_{\text{norm}}$.*

The qualitative statement being put forth by a *universal approximation theorem* (e.g. Leshno et al. (1993); Petrushev (1999); Yarotsky (2017a); Suzuki (2019); Grigoryeva & Ortega (2019); Heinecke et al. (2020); Kidger & Lyons (2020); Zhou (2020); Kratsios & Bilokopytov (2020); Siegel & Xu (2020); Kratsios & Hyndman (2021); Kratsios et al. (2022); Yarotsky (2022)) is a statement about the topological genericness of a machine learning model, such as a neural network model, in specific sets topological "function" spaces. Topological genericness is called *denseness*, and we say that a subset $F \subseteq X$ is dense with respect to a topology $\tau_X$ on $X$ if: for every non-empty open subset $U \in \tau_X$ there exists an element $f \in F$ which also belongs to $U$.

Related is the notion of *convergence* of a sequence in a general topological space $X$. Let $I$ be a set with a preorder $\preccurlyeq$ (i.e. for every $i, j, k \in I$ $i \preccurlyeq i$ and if $i \preccurlyeq j$ and $j \preccurlyeq k$ then $i \preccurlyeq k$), such that every finite subset of $I$ has an upper-bound with respect to $\preccurlyeq$. A typical example of a directed set is $\mathbb{N}$ equipped with the preorder given by $\leq$. A *net* in a topological space $X$ is a map from a directed set $I$ to $X$; we denote nets by $(x_i)_{i \in I}$. A typical example of a net is a sequence; in which case the directed set $I$ is the natural numbers with pre-order $\leq$. The next $(x_i)_{I \in I}$ is said to *converge* to an element $x$ of $X$ with respect to the topology $\tau_X$ if: for every $U \in \tau_X$ containing $x$, there exists some $i_U \in I$ such that for every $i \in I$ if $i_U \preccurlyeq I$ then $x_i \in U$.

## 2.4 Limit-Banach Spaces (LB-Spaces)

Our construction will exploit a special class of *topological vector spaces*, i.e. vector spaces wherein addition and scalar multiplication are continuous operators, formed by inductively *gluing* together ascending sequences of Banach spaces. Specifically, a topological vector space $X$ is a *limit-Banach* space, nearly always referred to as an LB-space in the literature, if first, one can exhibit sequence of strictly nested Banach spaces $\{X_n\}_{n=1}^{\infty}$ (i.e. each $X_n$ is a proper subspace of $X_{n+1}$) such that

$$X = \cup_{n=1}^{\infty} X_n.$$

Then, the topology on $X$ must be *smallest* topology containing every convex subset $B \subseteq X$ for which $kb \in B$ whenever $k \in [-1, 1]$ and $b \in B$, and for every positive integer $n$, $0 \in B \cap X_n$ and $B \cap X_n$ is an open subset of $X_n$.

Conversely, given a sequence of strictly nested Banach spaces $\{X_n\}_{n=1}^{\infty}$ one can always form an "optimal" LB-space as follows. Define $X \stackrel{\text{def.}}{=} \cup_{n=1}^{\infty} X_n$ and equip $X$ with the finest topology making $X$ into an LB-space and such that, for every $n \in \mathbb{N}_+$, the inclusion $X_n \subseteq X$ is continuous. Indeed, as discussed in (Osborne, 2014, Section 3.8), such a topology always exists[1]. We will henceforth refer to $X$ as the *LB-space glued together from* $\{X_n\}_{n=1}^{\infty}$.

A classical example of an LB-space arises when one wants to analyse polynomial functions but does not want to take their closure in some larger space (e.g. a larger space containing power series). We now present this example.

**Example 6** (Polynomial Functions). *For every $n \in \mathbb{N}_+$, the set of degree at-most $n$ polynomial functions mapping $\mathbb{R}$ to $\mathbb{R}$ is $\mathbb{R}_n[X] \stackrel{\text{def.}}{=} \{p(x) = \sum_{i=0}^{n} \beta_i x^i \beta \in \mathbb{R}^{n+1}\}$. We make $\mathbb{R}_n[X]$ into a Banach space through its identification the coefficients of polynomials in $\mathbb{R}_n[X]$ with $\mathbb{R}^{n+1}$; i.e. for any polynomial $p(x) = \sum_{i=0}^{n} \beta_i x^i \in \mathbb{R}_n[X]$ we define $\|p\|_n$ by*

$$\|p\|_n \stackrel{\text{def.}}{=} (\beta_0^2 + \cdots + \beta_n^2)^{1/2}. \tag{2}$$

*Thus, we may consider the $\mathbb{R}[X] \stackrel{\text{def.}}{=} \cup_{n=1}^{\infty} \mathbb{R}_n[X]$ to be the LB-space glued together from $\{\mathbb{R}_n[X]\}_{n=1}^{\infty}$ and $\mathbb{R}[X]$ consists precisely of all polynomial functions from $\mathbb{R}$ to $\mathbb{R}$ of any degree.*

*To illustrate the "optimality" of our LB-space, let us compare $\mathbb{R}[X]$ with the smallest Banach space containing every $\{\mathbb{R}_n[X]\}_{n=1}^{\infty}$ as a subspace. Notice that for every positive integer $n$, $\mathbb{R}_n[X]$ is a subspace of the following Hilbert space of (formal) power-series $\mathbb{R}[[X]] \stackrel{\text{def.}}{=} \{f(x) = \sum_{i=0}^{\infty} \beta_i x^i : \sum_{i=0}^{\infty} \beta_i^2 < \infty\}$ mapping $\mathbb{R}$ to $[-\infty, \infty]$ and normed by*

$$\|f\|_{\infty} \stackrel{\text{def.}}{=} \left(\sum_{i=0}^{\infty} \beta_i^2\right)^{1/2}.$$

---

[1]In the language of category theory, $X$ is the colimit of the inductive system $(\{X_n\}_{n=1}^{\infty}, \subseteq)$ in the category of locally-convex topological vector spaces with bounded linear maps as morphisms.

*By construction $\mathbb{R}[X]$ does not contain any function of the form $f(x) = \sum_{i=1}^{\infty} \beta_i x^i$ where an infinite number of $\beta_i$ are equal to zero while $\mathbb{R}[[X]]$ does contain such functions; e.g. $f(x) \stackrel{\text{def.}}{=} \sum_{i=0}^{\infty} \frac{x^i}{2^i}$ belongs to $\mathbb{R}[[X]]$ but not to $\mathbb{R}[X]$. In this way, the topological vector space $\mathbb{R}[X]$ is smaller than $\mathbb{R}[[X]]$ precisely because its topology is stronger.*

*Let us illustrate the topology on the LB-space $\mathbb{R}[X]$. By (Osborne, 2014, Proposition 3.40) we know that a convex subset $U \subseteq \mathbb{R}[X]$ is open if and only if $U \cap \mathbb{R}_n[X]$ is open for the topology on $\mathbb{R}_n[X]$ defined by the norm in equation 2.*

Example 6 illustrates the intuition behind *LB-spaces glued together from* $\{X_n\}_{n=1}^{\infty}$; namely, these spaces are "minimal limits" of sequences Banach spaces which contain no new element not already present in the Banach spaces $\{X_n\}_{n=1}^{\infty}$.

# 3  The CSL$^1$-Topology $\tau$

We now construct the csL$^1$-topology $\tau$ of Theorem 2 on the set $L^1_{\mu,\text{loc}}(\mathbb{R}^d, \mathbb{R}^D)$, in three steps. However, before beginning our construction, we fix an arbitrary "good a.e. partition" of $\mathbb{R}^d$. As we will see shortly, the construction of the csL$^1$-topology $\tau$ is independent of the choice of "good a.e. partition" of $\mathbb{R}^d$; and thus, the construction is natural (in the precise algebraic sense describe in Proposition 3, below). However, to establish this surprising algebraic property of the csL$^1$-topology $\tau$, it is more convenient to describe the construction (for any arbitrary choice of $\{K_n\}_{n=1}^{\infty}$) once and for all.

**Definition 1** (Good a.e. partition of $\mathbb{R}^d$). *A collection $\{K_n\}_{n=1}^{\infty}$ of compact subsets of $\mathbb{R}^d$ is called a good a.e. partition if it satisfies the following conditions:*

*(i) The set $\mathbb{R}^d - \cup_{n=1}^{\infty} K_n$ has Lebesgue measure $0$,*

*(ii) For every $n \in \mathbb{N}_+$, $K_n$ has positive Lebesgue measure,*

*(iii) For each $n, m \in \mathbb{N}_+$, if $n \neq m$ then $K_n \cap K_m$ has Lebesgue measure $0$.*

For instance, since our construction will be shown to be *independent of our choice* of a good a.e. partition of $\mathbb{R}^d$ made when constructing $\tau$. Once we show this, we may, without loss of generality, henceforth only consider the following partition of $\mathbb{R}^d$; illustrate in Figure 2.

**Example 7** (Good a.e. partition into Cubic Annuli). *For each $n \in \mathbb{N}_+$ set $K_n \stackrel{\text{def.}}{=} \{x \in \mathbb{R}^d : n < \|x\|_{\infty} \leq n+1\}$, where $\|x\|_{\infty} \stackrel{\text{def.}}{=} \max_{i=1,\dots,n} |x_i|$. Then $\{K_n\}_{n=1}^{\infty}$ is a good a.e. partition of $\mathbb{R}^d$.*

Let us construct the csL$^1$-topology $\tau$, using a fixed good a.e. partition of $\mathbb{R}^d$ in three steps.
**Step 1:** Given $\{K_n\}_{n=1}^{\infty}$ a good a.e. partition of $\mathbb{R}^d$ define the strictly nested sequence of Banach subspaces of $L^1(\mathbb{R}^d, \mathbb{R}^D)$ as follows. For every $n \in \mathbb{N}_+$ let $L^1_n(\mathbb{R}^d, \mathbb{R}^D)$ consist of all $f \in L^1(\mathbb{R}^d, \mathbb{R}^D)$ with ess-supp$(f) \subseteq \cup_{i=1}^n K_i$.
**Step 2:** The spaces $\{L^1_n(\mathbb{R}^d, \mathbb{R}^D)\}_{n=1}^{\infty}$ are aggregated into one LB-space, denoted by $L^1_c(\mathbb{R}^d, \mathbb{R}^D)$, whose underlying set is $\bigcup_{n \in \mathbb{N}^+} L^1_n(\mathbb{R}^d, \mathbb{R}^D)$ and equipped with the *finest topology* ensuring that the inclusions $L^1_n(\mathbb{R}^d, \mathbb{R}^D) \subseteq L^1_c(\mathbb{R}^d, \mathbb{R}^D)$ remain continuous.

**Remark 1** (Notation and Independence of Choice of Good a.e. Partition of $\mathbb{R}^d$). *The notation $L^1_c(\mathbb{R}^d, \mathbb{R}^D)$ does not make any reference to our choice of a good a.e. partition of $\mathbb{R}^d$ used to define the space $L^1_c(\mathbb{R}^d, \mathbb{R}^D)$. This is because, as we will shortly see in Proposition 3 below, the topology on $L^1_c(\mathbb{R}^d, \mathbb{R}^D)$ is independent of our choice of a good a.e. partition of $\mathbb{R}^d$ used to define it. However, to formally state that result; we will make use of the notation $L^1_c(\{K_n\}_{n=1}^{\infty}, \mathbb{R}^D)$ emphasizing our choice of $\{K_n\}_{n=1}^{\infty}$ which is a good a.e. partition of $\mathbb{R}^d$ used in Steps 1 and 2.*

**Step 3:** Since $L^1_c(\mathbb{R}^d, \mathbb{R}^D)$ does not contain every function in $L^1_{\text{loc}}(\mathbb{R}^d, \mathbb{R}^D)$ then, intuitively speaking, we "glue" remaining locally-integrable functions to $L^1_c(\mathbb{R}^d, \mathbb{R}^D)$ by aggregating the topologies on $L^1(\mathbb{R}^d, \mathbb{R}^D)$ and on $L^1_{\text{loc}}(\mathbb{R}^d, \mathbb{R}^D)$ to $L^1_c(\mathbb{R}^d, \mathbb{R}^D)$. Rigorously, we define this gluing as follows.

**Definition 2** (CSL$^1$-Topology $\tau$). *The csL$^1$-topology $\tau$ on $L^1_{\mu,loc}(\mathbb{R}^d, \mathbb{R}^D)$ is smallest[2] topology on $L^1_{\text{loc}}(\mathbb{R}^d, \mathbb{R}^D)$ containing $\tau_c \cup \tau_{\text{norm}} \cup \tau_{\text{loc}}$.*

---

[2]I.e. $\tau_c \cup \tau_{\text{norm}} \cup \tau_{\text{loc}}$ is a subbase for the topology $\tau$.

Since $\tau_{\mathrm{norm}}$, $\tau_{\mathrm{loc}}$, and $\tau_c$ all exist and since the smallest topology containing a collection of sets[3] must exist (see (Munkres, 2000, page 82)); thus, $\tau$ exists. Next, we examine the key properties of $\tau$ for our problem. Namely, how it compares to the usual topologies on $L^1_{\mathrm{loc}}(\mathbb{R}^d, \mathbb{R}^D)$ and on $L^1(\mathbb{R}^d, \mathbb{R}^D)$, as well as its independence of the choice of good a.e. partition of $\mathbb{R}^d$ used to construct it.

### 3.1 Properties of the CSL[1]-Topology $\tau$

It is straightforward to see that any $f \in L^1_{\mathrm{loc}}(\mathbb{R}^d, \mathbb{R}^D)$ which is essential supported on some $\cup_{i=1}^n K_i$ for some $n \in \mathbb{N}_+$ belongs to $L^1_c(\mathbb{R}^d, \mathbb{R}^D)$. However, Proposition 3 below implies that every essentially compactly supported Lebesgue-integrable functions must belong to $L^1_c(\mathbb{R}^d, \mathbb{R}^D)$ since the set $L^1_c(\mathbb{R}^d, \mathbb{R}^D)$ and its topology are both independent of the choice of good a.e. partition $\{K_n\}_{n=1}^\infty$ of $\mathbb{R}^d$ used to construct $L^1_c(\mathbb{R}^d, \mathbb{R}^D)$.

The result also points to the naturality of the csL[1]-topology $\tau$'s construction. By which we mean that $\tau$ has the surprising and convenient algebraic property it is independent of the good a.e. partition used to build it.

**Proposition 3** (The csL[1]-topology $\tau$ is independent of the choice of good a.e. partition). *Let $\{K_n\}_{n=1}^\infty$ and $\{K'_n\}_{n=1}^\infty$ be good a.e. partitions of $\mathbb{R}^d$. Then $L^1_c(\{K_n\}_{n=1}^\infty, \mathbb{R}^D) = L^1_c(\{K'_n\}_{n=1}^\infty, \mathbb{R}^D)$. Consequentially, $\tau$ is independent of the good a.e. partition of $\mathbb{R}^d$ used to construct it.*

The significance of Proposition 3 is that it allows us to reduce our entire understanding of the problem, and many of our proofs, to simply considering a single "canonical" good a.e. partition of $\mathbb{R}^d$ which is easy to work with; namely, the Cubic Annuli of Example 7. Briefly, the reason for this is that, given a good a.e. partition of $\mathbb{R}^d$ $\{K_n\}_{n=1}^\infty$, the approximation of a compactly supported Lipschitz function $f : \mathbb{R}^d \to \mathbb{R}^D$ in $\tau$ requires us to identify the smallest $n \in \mathbb{N}_+$ for which we can identify its support with respect to $\{K_n\}_{n=1}^\infty$ which we use to discretize $\mathbb{R}^d$; i.e.

$$\text{ess-supp}(\hat{f}) \subseteq \cup_{i=1}^n K_i. \tag{3}$$

Then, we must approximate $\hat{f}$ in the $L^1$-norm on $\cup_{i=1}^{n+1} K_i$ using our model. The intuitive message of Proposition 3 is that, given any other good a.e. partition of $\mathbb{R}^d$ $\{K'_n\}_{n=1}^\infty$, the compactness of $\cup_{i=1}^{n+1} K_i$ implies that there is a smallest $n_1 \in \mathbb{N}_+$ such that $\cup_{i=1}^n K_i \subseteq \cup_{j=1}^{n_1} K'_j$ thus, there must be a smallest integer for which equation 3 holds with $\{K'_n\}_{n=1}^\infty$ holds in place of $\{K_n\}_{n=1}^\infty$. To see the equivalence, arguing similarly, there must exist an (other) $n_2 \in \mathbb{N}_+$ such that $\cup_{j=1}^{n_1} K'_j \subseteq \cup_{i=1}^{n_2} K_i$. Therefore, we may interchangeable identify where the support of a compactly supported Lipschitz function lies using any discretization of $\mathbb{R}^d$ by any choice of good a.e. partition of $\mathbb{R}^d$.

The next result shows that the csL[1]-topology $\tau$ on $L_{\mathrm{loc}}(\mathbb{R}^d, \mathbb{R}^D)$ is strictly finer than the norm metric topology thereon, and its restriction to $L^1(\mathbb{R}^d, \mathbb{R}^D)$ is strictly stronger than the norm topology thereon ((Nagata, 1974, Chapter 2.4)). The approximation-theoretic implication is that fewer members of $L_{\mathrm{loc}}(\mathbb{R}^d, \mathbb{R}^D)$ can be approximated by deep learning models in $\tau$ than in the other two topologies.

**Proposition 4.** *The csL[1]-topology $\tau$ is strictly stronger than $\tau_{\mathrm{loc}}$.*

The phenomenon of Proposition 4 persists when restricting the csL[1]-topology $\tau$ to the subset $L^1(\mathbb{R}^d, \mathbb{R}^D)$ of $L^1_{\mathrm{loc}}(\mathbb{R}^d, \mathbb{R}^D)$ and comparing it with the norm topology (which is stronger than $\tau_{\mathrm{loc}}$ restricted to $L^1(\mathbb{R}^d, \mathbb{R}^D)$).

**Proposition 5.** *The restriction of the csL[1]-topology $\tau$ to $L^1(\mathbb{R}^d, \mathbb{R}^D)$ is strictly stronger than the norm topology $\tau_{\mathrm{norm}}$ on $L^1(\mathbb{R}^d, \mathbb{R}^d)$.*

We are now in a position to prove Theorem 2. The next section outlines the main steps in the theorem's derivation, with the details being relegated to our paper's appendix.

## 4 Outline of the Proof of The Main Results

To better understand our main results we overview the principal steps undertaken in their derivation. We begin by establishing the universality of $\mathrm{NN}^{\mathrm{ReLU+Pool}}$ for the cs$L^1$-topology, as guaranteed by Theorem 2. We build up the properties of the csL[1]-topology $\tau$ along the way and we use them to derive the aforementioned results; whereby deriving Theorem 1 and Theorem 3 along the way. Propositions 1 and 2 are derived at the end.

---

[3]Given a set $X$ and a collection of subsets $A$ of $X$, the smallest topology $\tau_A$ on $X$ containing a $A$ is called the topology generated by $A$ and $A$ is called a subbase of $\tau_A$.

### 4.1 Establishing Theorems 2 and 3: The universality of $\mathrm{NN}^{\mathrm{ReLU+Pool}}$ in the topology $\tau$

In order to establish Theorem 2, we must first understand how density in $L^1_{\mathrm{loc}}(\mathbb{R}^d, \mathbb{R}^D)$, for the metric topology interacts with density in $L^1_{\mathrm{loc}}(\mathbb{R}^d, \mathbb{R}^D)$ for the $\mathrm{cs}L^1$-topology. The next lemma accomplishes precisely this, by showing how dense subsets of $L^1_{\mathrm{loc}}(\mathbb{R}^d, \mathbb{R}^D)$ for the metric topology can be used to construct dense subsets of $L^1_{\mathrm{loc}}(\mathbb{R}^d, \mathbb{R}^D)$ for the $\mathrm{cs}L^1$-topology. This construction happens in two phases. First, each "function" in the original dense subset is localized so that it is essentially supported on a part $K_n$ in (any) good a.e. partition $\{K_n\}_{n=1}^\infty$ of $\mathbb{R}^d$. Then, each of these localized "functions" are then pieced back together to form a new "function" which is essentially supported on the compact subset $\cup_{i=1}^n K_n$.

Let $\mathrm{Lip}_c(\mathbb{R}^d, \mathbb{R}^D)$ denote the set of "compact support" Lipschitz functions $f : \mathbb{R}^d \to \mathbb{R}^D$; i.e. $f$ is Lipschitz and $\mathrm{ess\text{-}supp}(f)$ is a compact subset of $\mathbb{R}^d$. The first key observation in the proof of Theorem 2 is that, $\mathrm{Lip}_c(\mathbb{R}^d, \mathbb{R}^D)$ is dense in $L^1_{\mathrm{loc}}(\mathbb{R}^d, \mathbb{R}^D)$ for the $\mathrm{cs}L^1$-topology $\tau$.

**Lemma 1** (Density of compactly-supported Lipschitz functions in the $\mathrm{cs}L^1$-topology $\tau$). *The set $\mathrm{Lip}_c(\mathbb{R}^d, \mathbb{R}^D)$ is dense in $L^1_{\mathrm{loc}}(\mathbb{R}^d, \mathbb{R}^D)$ for the $\mathrm{cs}L^1$-topology $\tau$.*

The second key observation, also contained in the next lemma, is a sufficient condition for approximating a "compact support" Lipschitz function with respect to the $\mathrm{cs}L^1$-topology $\tau$. Briefly, the approximation of such a function in $\tau$ involves the simultaneous approximation of its *outputs* as well as its *essential support*.

**Lemma 2** (Approximation of compactly-supported Lipschitz functions in the $\mathrm{cs}L^1$-topology $\tau$). *Let $f \in L^1(\mathbb{R}^d, \mathbb{R}^D)$ be Lipschitz and $\mathrm{ess\text{-}supp}(f)$ be compact, $\{K_n\}_{n=1}^\infty$ be the cubic-annuli of Example 7. If $\{f_n\}_{n=1}^\infty$ is a sequence in $L^1_{\mathrm{loc}}(\mathbb{R}^d, \mathbb{R}^D)$ for which there is an $n_f \in \mathbb{N}_+$ with*

$$\lim_{n\uparrow\infty} \|f_n - f\|_{L^1(\mathbb{R}^d, \mathbb{R}^D)} = 0 \ and \ \mathrm{ess\text{-}supp}(f) \cup \bigcup_{n=1}^\infty \mathrm{ess\text{-}supp}(f_n) \subseteq [-n_f - 1, n_f + 1]^d, \tag{4}$$

*then $\{f_n\}_{n=1}^\infty$ converges to $f$ in the $\mathrm{cs}L^1$-topology $\tau$.*

Together, Lemmata 2 and 1 provide a sufficient condition for universality with respect to the $\mathrm{cs}L^1$-topology. Furthermore the condition is in a sense quantitative. We say in a sense, since the topology $\tau_c$ is non-metrizable (see (Narayanaswami & Saxon, 1986, Corollary 3) and consequentially $\tau$ is non-metrizable); thus there is no metric describing the approximation of a function in $\tau$. I.e. no genuine quantitative statement is possible[4]. The next lemma, Proposition 3, and Example 7 form the content of Theorem 1 (ii).

**Lemma 3** (Approximation of a compactly essentially-supported functions in the $\mathrm{cs}L^1$-topology $\tau$). *Let $\mathscr{F} \subseteq L^1_{\mathrm{loc}}(\mathbb{R}^d, \mathbb{R}^D)$. If for every $f \in \mathrm{Lip}_c(\mathbb{R}^d, \mathbb{R}^D)$ there exists a sequence $\{f_n\}_{n=1}^\infty$ in $\mathscr{F}$ satisfying the condition equation 4 then, $\mathscr{F}$ is dense in $L^1_{\mathrm{loc}}(\mathbb{R}^d, \mathbb{R}^D)$ for the $\mathrm{cs}L^1$-topology $\tau$.*

By Lemma 3, it therefore remains to construct a subset of networks in $\mathrm{NN}^{\mathrm{ReLU+Pool}}$ which can approximate any compactly supported Lipschitz function in the $L^1$-norm and simultaneously correctly identify its essential support via the cubic annuli partition of $\mathbb{R}^d$. Figure 1 illustrates the main points of the next lemma; namely, if the target function is compactly supported then its output can be closely approximated by a ReLU network which also simultaneously correctly identifies the integer $n$ such that the target function is supported in the $d$-dimensional cube $[-n-1, n+1]^d$.

Accordingly, our next lemma is an extension of the main theorem of Shen et al. (2022), which gives an estimate on the width and depth of the smallest deep ReLU network approximating a Lipschitz map from a compact subset $X$ of $\mathbb{R}^d$ to $\mathbb{R}^D$ (instead of the case where $D = 1$ and $X = [0, 1]^d$).

**Lemma 4** (Uniform approximation of Lipschitz maps on low-dimensional compact subsets of $\mathbb{R}^d$). *Let $X \subseteq \mathbb{R}^d$ be non-empty and compact and let $f : X \to \mathbb{R}^D$ be Lipschitz. For every "depth parameter" $L \in \mathbb{N}_+$ and "width parameter" $N \in \mathbb{N}_+$ there exists a $\hat{f} \in \mathrm{NN}^{\mathrm{ReLU}}$ satisfying the uniform estimate*

$$\max_{x \in X} \|f(x) - \hat{f}(x)\| \lesssim \log_2(\mathrm{cap}(X)) \, \mathrm{diam}(X) \, \mathrm{Lip}(f) \, \frac{D^{3/2} d^{1/2}}{N^{2/d} L^{2/d} \log_3(N+2)^{1/d}},$$

---

[4]Another example of a non-metric universal approximation theorem in the deep learning literature is the universal classification result of (Kratsios & Bilokopytov, 2020, Corollary 3.12)).

*where $\lesssim$ hides an absolute positive constant independent of $X, d, D,$ and $f$. Furthermore, $\hat{f}$ satisfies*

1. **Width:** *$\hat{f}$'s width is at-most $d(D+1) + 3^{d+3} \max\{d\lfloor N^{1/d}\rfloor, N+2\}$*

2. **Depth:** *$\hat{f}$'s depth is at-most $D(11L + 2d + 19)$.*

In order to apply Lemma 4, we need our approximating model to have support which "matches" the support of the target function $f \in L^1_c(\mathbb{R}^d, \mathbb{R}^D) \stackrel{\text{def.}}{=} \bigcup_{n \in \mathbb{N}^+} L^1_n(\mathbb{R}^d, \mathbb{R}^D)$ being approximated. The next lemma describes how, given a ReLU network how one can build a new ReLU network with one pooling layer at its output, which coincides with the original network on an arbitrarily cubic-annuli (as in Example 7) and vanishes straightaway outsides the correct number of cubic-annuli (with possibly one extra part of the good a.e. partition of $\mathbb{R}^d$).

**Lemma 5** (Adjusting a ReLU network to have support on the union of the first $n+1$ cubic annuli)**.**
*Let $\log_2(d) \in \mathbb{N}_+$ and $\hat{f} \in \mathrm{NN}^{\mathrm{ReLU}}$ have depth $d_{\hat{f}}$ and width $w_{\hat{f}}$. For every $n \in \mathbb{N}_+$ and each $0 < \delta < 1$, there exists a $\hat{f}^{\mathrm{pool}} \in \mathrm{NN}^{\mathrm{ReLU+Pool}}$ with width $\max\{d(d-1) + 2, D\} + w_{\hat{f}}$ and depth $2 + 3d + d_{\hat{f}}$ satisfying:*

(i) **Implementation on the Cube:** *For each $x \in [-n, n]^d$ it holds that $\hat{f}(x) = \hat{f}^{\mathrm{pool}}(x)$,*

(ii) **Controlled Support:** *$\mathrm{ess\text{-}supp}(\hat{f}) \subseteq \left[-\sqrt[d]{2^{-d}\varepsilon + n^d}, \sqrt[d]{2^{-d}\varepsilon + n^d}\right]^d$,*

(i) **Control of Error Near the Boundary:** *$\left\|\hat{f} - \hat{f}^{\mathrm{pool}}\right\|_{L^1(\mathbb{R}^d, \mathbb{R}^D)} < \varepsilon$.*

Lemmata 1, 2, and 3 imply that $\mathrm{NN}^{\mathrm{ReLU+Pool}}$ is dense in $L^1_{\mathrm{loc}}(\mathbb{R}^d, \mathbb{R}^D)$ for the csL$^1$-topology $\tau$ only if $\mathrm{NN}^{\mathrm{ReLU+Pool}}$ has a subset which can approximate any essentially compactly-supported Lipschitz function while having almost correct support (as detected by the cubic-annuli partition) as formalized by condition 4. Since Lemma 5 implies that such a subset of networks in $\mathrm{NN}^{\mathrm{ReLU+Pool}}$ exists then, Theorem 2 follows.

*Proof of Theorem 2.* The result for PW-Lin $=$ ReLU is a direct consequence of Lemmata 4 and 5 applied to Lemma 3. The result for general non-affine piecewise linear activation functions from the ReLU case by (Yarotsky, 2017b, Proposition 1). This is because (Yarotsky, 2017b, Proposition 1) states that any network in $\mathrm{NN}^{\sigma_{\mathrm{PW-Lin}}}$ can be implemented by a network in $\mathrm{NN}^{\mathrm{ReLU}}$. □

We are now equally in a position to prove the first claim in theorem Theorem 3.

*Proof of Theorem 3.* Since $f$ is compactly essentially-supported, by Lemma 4 there is an $\hat{f}^{\varepsilon_n/2} \in \mathrm{NN}^{\mathrm{ReLU}}$ satisfying

$$\max_{x \in \mathrm{ess\text{-}sup}(f)} \left\|f(x) - \hat{f}^{\varepsilon_n/2}(x)\right\| < \frac{\varepsilon_n}{2}, \tag{5}$$

with width $w_{\hat{f}^{\varepsilon_n/2}}$ at-most $d(D+1) + 3^{d+3}\max\{d\lfloor N^{1/d}\rfloor, N+1\}$ and depth $d_{\hat{f}^{\varepsilon_n/2}}$ equal to

$$d_{\hat{f}^{\varepsilon_n/2}} \stackrel{\text{def.}}{=} \frac{\varepsilon_n^{-d/2}}{N\log_3(N+2)^{1/2}}\left(2\log_2(\mathrm{cap}(\mathrm{ess\text{-}supp}(f)))\,\mathrm{diam}(\mathrm{ess\text{-}supp}(f))\,\mathrm{Lip}(f)\right)^d (cD^{3/d}d^d), \tag{6}$$

where $c > 0$ is an absolute constant independent of $X, d, D,$ and $f$. Set $n_f \stackrel{\text{def.}}{=} \min\{n \in \mathbb{N}_+ : \mathrm{ess\text{-}supp}(f) \subseteq [-n,n]^d\}$ and apply Lemma 5 to $\hat{f}^{\varepsilon_n/2}$ there exists an $\hat{f}^{(n)} \in \mathrm{NN}^{\mathrm{ReLU+Pool}}$ with

$\mathrm{ess\text{-}supp}(\hat{f}^{(n)}) \subseteq \left[-\sqrt[d]{2^{-d-1}\varepsilon_n + n_f^d}, \sqrt[d]{2^{-d-1}\varepsilon_n + n_f^d}\right]^d$, equal to $\hat{f}^{\varepsilon_n/2}$ on $[-n_f, n_f]^d$ and such that $\left\|\hat{f}^{(n)} - \hat{f}^{\mathrm{pool}}\right\|_{L^1(\mathbb{R}^d, \mathbb{R}^D)} < \frac{\varepsilon_n}{2}$. Therefore, the estimate in equation 5 and implies that

$$\max_{x \in \mathrm{ess\text{-}sup}(f)} \left\|f(x) - \hat{f}^{(n)}(x)\right\| \leq \max_{x \in \mathrm{ess\text{-}sup}(f)} \left\|f(x) - \hat{f}^{(n)}(x)\right\| + \max_{x \in \mathrm{ess\text{-}sup}(f)} \left\|\hat{f}^{(n)}(x) - \hat{f}^{\varepsilon_n/2}(x)\right\| \leq 2^{-1}\varepsilon_n + 2^{-1}\varepsilon_n = \varepsilon_n.$$

Similarly, equation 5 implies that

$$\|f - \hat{f}^{(n)}\|_{L^1} \leq \|f - \hat{f}^{\varepsilon_n/2}\|_{L^1} + \|\hat{f}^{(n)} - \hat{f}^{\varepsilon_n/2}\|_{L^1}$$

and that both $\hat{f}^{(n)}$ and $f$ are essentially-supported in $[-n_f - 1, n_f + 1]^d$; whence, for each $n \in \mathbb{N}_+$ the condition equation 4 is met. Therefore, Lemma 2 implies that the sequence $\{\hat{f}^{(n)}\}_{n=1}^{\infty}$ in $\mathrm{NN}^{\mathrm{ReLU+Pool}}$ converges to $f$ in the csL$^1$-topology $\tau$.

It remains to count each of $\hat{f}^{(n)}$'s parameters. By construction, Lemma 5 and the estimate on $w_{\hat{f}^{\varepsilon/2}}$ (below equation 5) imply that $\hat{f}^{(n)}$ has width at-most $\max\{d(d-1)+2, D\} + d(D+1) + 3^{d+3} \max\{d\lfloor N^{1/d}\rfloor, N+1\}$. Similarly, Lemma 5 and equation 6 imply that each $\hat{f}^{(n)}$ has depth equal to

$$\frac{\varepsilon_n^{-d/2}}{N \log_3(N+2)^{1/2}} \left( \log_2(\mathrm{cap}(\mathrm{ess\text{-}supp}(f))) \, \mathrm{diam}(\mathrm{ess\text{-}supp}(f)) \, \mathrm{Lip}(f) \right)^d (c \, 2^d D^{3/d} d^d + 3d) + 2d + 2.$$

Relabeling $C_1 \stackrel{\text{def.}}{=} c \, 2^d D^{3/d} d^d + 3d$, $C_2 \stackrel{\text{def.}}{=} +2d + 2$, $C_3 \stackrel{\text{def.}}{=} \max\{d(d-1)+2, D\}$, $C_4 \stackrel{\text{def.}}{=} d(D+1) + 3^{d+3}$, yields the first conclusion. $\qquad\square$

## 4.2 Establishing Propositions 1 and 2: The Non-University of Analytic Models in the Topology $\tau$

The main step in showing that $\mathrm{NN}^{\sigma}$ fails to be dense in $L^1_{loc}(\mathbb{R}^d, \mathbb{R}^D)$ for the csL$^1$-topology is the following *necessary condition* for a sequence $\{f_n\}_{n=1}^{\infty}$ in $L^1_{loc}(\mathbb{R}^d, \mathbb{R}^D)$ to convergence to some essentially compactly supported $f \in L^1_{loc}(\mathbb{R}^d, \mathbb{R}^D)$ therein with respect to $\tau$. Moreover, Proposition 3, and Example 7 Theorem 1 (i).

**Proposition 6** (Necessary condition for convergence in the csL$^1$-topology $\tau$). *Let $n \in \mathbb{N}_+$ and $f \in L^1_n(\mathbb{R}^d, \mathbb{R}^D)$. A sequence $\{f_k\}_{k \in \mathbb{N}^+}$ in $L^1_{loc}(\mathbb{R}^d, \mathbb{R}^D)$ converges to $f$ with respect to the csL$^1$-topology $\tau$, only if there is some $N \in \mathbb{N}_+$ with $N \geq n$ such that all but a finite number of $f_k$ are in $L^1_N(\mathbb{R}^d, \mathbb{R}^D)$ and $\lim_{k \uparrow \infty} \|f_k - f\| = 0$.*

Together, Proposition 6 and the fact that if any analytic function is 0 on a non-empty open subset of $\mathbb{R}^d$ then it must be identically 0 everywhere on $\mathbb{R}^d$ (see (Griffiths & Harris, 1994, page 1)) imply that no analytic function can converge to an essentially compactly supported "function" in $L^1_{loc}(\mathbb{R}^d, \mathbb{R}^D)$ with respect to the cs$L^1$-topology.

**Lemma 6** (Families of analytic functions cannot be dense with respect to the csL$^1$-topology $\tau$). *If $\mathscr{F}$ is a set of analytic functions from $\mathbb{R}^d$ to $\mathbb{R}^D$ then*

1. *$\mathscr{F}$ is not dense in $L^1_{loc}(\mathbb{R}^d, \mathbb{R}^D)$ for the csL$^1$-topology $\tau$.*

2. *If $f : \mathbb{R}^d \to \mathbb{R}^D$ is Lipschitz, is compact essential-supported, and not identically 0 then, is a sequence $\{\varepsilon_n\}_{n=1}^{\infty}$ in $(0, \infty)$ converging to 0 such that no $\hat{f} \in \mathscr{F}$ satisfies both Theorem 3 (i) and (iii).*

The proof of Theorem 2 (ii) is a consequence of Lemma 6 and the observation that any network in $\mathrm{NN}^{\omega+\mathrm{Pool}}$ is an analytic function.

*Proof of Theorem 2 (ii).* By Lemma 6, the class of analytic functions from $\mathbb{R}^d$ to $\mathbb{R}^D$, denoted by $C^{\omega}(\mathbb{R}^d, \mathbb{R}^D)$, is not dense in $L^1_{loc}(\mathbb{R}^d, \mathbb{R}^D)$ for the cs$L^1$-topology. Now, the composition and the addition of analytic functions is again analytic. Since every affine function is analytic and since every activation function $\sigma \in C^{\omega}(\mathbb{R})$ is by definition analytic then, every $f \in \mathrm{NN}^{\omega}$ must be analytic. I.e, $\mathrm{NN}^{\omega} \subseteq C^{\omega}(\mathbb{R}^d, \mathbb{R}^D)$. Therefore, $\mathrm{NN}^{\omega}$ cannot be in $L^1_{loc}(\mathbb{R}^d, \mathbb{R}^D)$ for the cs$L^1$-topology. $\qquad\square$

The proof of Proposition 1 now also follows from Lemma 6.

*Proof of Proposition 1.* Since every polynomial function is analytic then, the result follows from Lemma 6. $\qquad\square$

*Proof of Theorem 3 (Continued).* If $f : \mathbb{R}^d \to \mathbb{R}^D$ is Lipschitz, compactly-supported, and not identically 0 then Lemma 6 and the fact that every $\hat{f} \in \text{NN}^{\omega + \text{Pool}} \cup \mathbb{R}[x_1, \ldots, x_d]$ is an analytic function implies that Theorem 3 (i)-(iii) cannot all hold simultaneously. This completes the proof of Theorem 3. □

We now discuss some technical points surrounding our results, a few of the implications of our findings, and how our analysis could be used to obtain similar constructions for networks designed to approximate solutions to PDEs.

## 5 Discussion

There are a few question which arise during our analysis which we now take the time to discuss. These are: "Is Theorem 2 about a refinement of the topology on $L^1_{\text{loc}}(\mathbb{R}^d, \mathbb{R}^D)$ in which $\text{NN}^{\text{ReLU} + \text{Pool}}$ is universal while $\text{NN}^{\omega + \text{Pool}}$ is not?", "Are ReLU networks better than networks with analytic activation functions?" and "What is the significance of the bilinear pooling later pool?"

### 5.1 Are ReLU Networks Better Than Networks With Analytic Activation Functions?

There are several explanations for a learning model's success over its alternatives for a *given learning task*. Some of the principle reasons for a model's successful inductive bias are its expressiveness, its ability to generalize well on a given type of problem, and how training dynamics interact with these two properties for a given problem (e.g. the impage of using different initialization schemes as studied by (Martens et al., 2021)).

A key point which we emphasize is that, the type of problem which we have implicitly considered in this paper concerns the approximation of a compactly supported function's output and its support simultaneously. Therefore, we ask the following question from the approximation-theoretic vantage point:

*"Are ReLU networks better than networks with analytic activation functions?"*

As one may expect, the answer is a mixed *"yes and no"*. Let us begin with "no" part of our answer to this question. If that is the task is to learn a solution to a PDE (e.g. Han et al. (2018); Beck et al. (2021a;b) physics-informed neural networks Raissi et al. (2019); Shin et al. (2020); Mishra & Molinaro (2021)). Then, the networks should exhibit non-trivial (higher-order) partial derivatives, and the approximation should be in the $C^k$-norm (for some $k > 0$). In such cases, it is known that ReLU networks are less effective than sigmoid, tanh, or SIREN networks; see Markidis (2021) or Hornik et al. (1990); Siegel & Xu (2020); De Ryck et al. (2021). A fortiori, it is rather straightforward to see this when $k \geq 2$ and $d = D = 1$, since any weak derivative of a ReLU neural network must vanish outside of a set of Lebesgue measure 0. This is the *"no"* part of the answer to the above question.

For the *"yes"* part of the answer, Theorem 2 implies that deep ReLU networks with bilinear pooling layer can approximate locally-integrable functions while exactly implementing their support (up to a good a.e. partition of $\mathbb{R}^d$). In contrast, as shown in Proposition 2, neural networks with analytic activation function cannot do this by virtue of their analyticity. Therefore, ReLU neural networks can be more suitable for learning tasks where the target function is known to be compactly supported.

### 5.2 What Is The Significance Of The Bilinear Pooling Layer Pool?

We conclude our discussion by considering one last question:

*"What is the significance of the bilinear pooling layer?"*

Our construction of a network $\hat{f} \in \text{NN}^{\text{ReLU} + \text{Pool}}$ realizing the conclusion of Theorem 3 for a given approximation error $\varepsilon > 0$ relies two distinct ReLU networks which are multiplied together using bilinear pooling layers. Suppose that $f : \mathbb{R}^d \to \mathbb{R}^D$ is a compactly supported Lipschitz function and let $n$ be the smallest integer for which ess-supp$(f)$ is contained in the union of the first $n$ Cubic Annuli of Example 7. The role first ReLU network $\hat{f}_{\text{mask}} : \mathbb{R}^d \to \mathbb{R}$ is to implements a piece-wise affine "mask" which takes values 0 outside of $\cup_{i=1}^{n+1} K_i$, value 1 in $\cup_{i=1}^{n} K_i$, and intermediate

value in $K_{n+1} - K_n$ just as in the construction of Yarotsky (2017b). The second ReLU network $\hat{f}^\varepsilon$ is constructed which approximates the target function $f : \mathbb{R}^d \to \mathbb{R}^D$ uniformly on the compact set ess-supp($f$) to $\varepsilon$-precision, and we construct the ReLU network $\hat{f}^\varepsilon$ in such a way that its depth and width depend on the dimension and metric capacity of ess-supp($f$) as well as on the regularity of the function $f$.

Lastly, using several bilinear pooling layers we construct the approximating network $\hat{f}$ in Theorem 3 which implements $\hat{f} = \hat{f}_{\text{mask}} \cdot \hat{f}^\varepsilon$. Consequentially, $\hat{f} \approx f$ for every $x \in$ ess-supp($f$) and it is supported exactly on $\cup_{i=1}^{n+1} K_i$ (i.e.: its support coincides with that of the target function up to our discretization of $\mathbb{R}^d$ as implemented by $\{K_n\}_{n=1}^\infty$). The subtle difference in our approach and in the constructions of Yarotsky (2017b); Kidger & Lyons (2020) is that those authors use small ReLU networks to *approximately implement* the multiplication operation $(x_1, x_2) \mapsto x_1 x_2$ instead of the bilinear pooling layers which we use. The issue here is that, their construction does not guarantee that an "approximate product" of $\hat{f}_{\text{mask}}$ and $\hat{f}^\varepsilon$ is supported in $\cup_{i=1}^{n+1} K_i$ nor that is has compact support; whence, there is no guarantee with that method that one can construct a deep ReLU network satisfying the conditions of Lemma 2. NB, this is not to say that a construction is impossible; but simply that it remains an *open question*.

## Conclusion

In this paper, we showed that deep feedforward networks with non-affine piecewise linear activation functions and bilinear pooling layers are approximation-theoretically well suited to tasks where the objective is to learn a compactly supported function; e.g. the distance map to a non-empty compact subset of Euclidean space. Theorem 1 translated this learning problem into a universal approximation problem by constructing a topology on the set $L^1_{\text{loc}}(\mathbb{R}^d, \mathbb{R}^D)$ of locally Lebesgue-integrable functions in which members of the subspace $L^1_c(\mathbb{R}^d, \mathbb{R}^D)$ of essentially compactly-supported integrable function could only be approximated by models which match their discretized support (as formalized by a good a.e. partition of $\mathbb{R}^d$).

Theorem 2 demonstrated that any feedforward neural network architecture with bilinear pooling and piecewise-linear (but non-affine) activation function is universal in this topological space. Consequentially showing that, ReLU networks with bilinear pooling layers are capable of approximating functions in $L^1_c(\mathbb{R}^d, \mathbb{R}^D)$ in $L^1$-norm while simultaneously implementing their support; up to a good a.e. partition of $\mathbb{R}^d$. Theorem 3 provided a quantitative and uniform refinement of this result for any compactly-supported Lipschitz function. The result also provided quantitative estimates on the width, depth, and the number of bilinear pooling layers required for a ReLU network to implement the said approximation. Moreover, our new proof techniques allowed us to explicitly encode the metric capacity and dimension of the target function's essential support into the models' complexity estimates.

## Funding

This first stage of this research was funded by the ETH Zürich Foundation (circa 2020-2021) and the second stage was funded by the European Research Council (ERC) Starting Grant 852821—SWING (during 2022).

## Acknowledgments

The authors would like to thank Luca Galimberti of NTNU, Florian Krach, Calypso Herrera, and Jakob Heiss from the ETH for their helpful feedback in the article's late stages. The authors would equally like to thank Ivan Dokmanić and Hieu Nguyen of the University of Basel for their helpful references concerning pooling layers and certain activation functions.

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
