# OpenReview forum: "Do ReLU Networks Have An Edge When Approximating Compactly-Supported Functions?"
_TMLR — Accepted by TMLR_

### Review · Reviewer_vPRx · 2022-05-11

**Summary Of Contributions:**

This paper theoretically studies the question of why NNs with piecewise linear (PW-Lin) activations like ReLU empirically outperform analytic activations (e.g. sigmoid). The authors attempt to answer this question by showing that piecewise linear activations are more expressive than  their analytic activation counterparts.

Given that available universal approximation theorems in the L1-norm apply for analytic activations, the key innovation here (imo) is to construct a strictly stronger topology on the space of (locally-integrable) functions than the topology induced by the L1- norm. With respect to their constructed topology, the authors show (Theorem 1) that the NNs with PW-Lin activations is dense in the space of locally integrable functions, but that NNs with analytic activations are not. The argument for the latter seems to rely on the fact that if an analytic function is 0 in a non-empty open susbet of $\mathbb{R}^d$ then it must be identically 0 (stated around Lemma 6).

The authors also show Theorem 3, which provides a quantitative bound on the necessary width and depth of an approximating NN with PW-Lin for a target function that is Lipschitz and compactly-supported.

**Requested Changes:**

Clarity:
1. Section 1 is quite confusing given that many terms e.g. $L_{\text{loc}}^1$ are not introduced until later e.g. Section 2/3. Also I'm not sure it makes sense to state all your results in section 1 especially because many of the terms are not defined. I would suggesting restructuring the paper to simply state the results in section 3.
2. In abstract, should $NN^{ReLU}$ also be +Pool?
3. The sentence 'At this point, we ask:...' comes out of nowhere and it is not very motivated the study of polynomial regressors in the build up to Theorem 2.
4. Theorem 3 ii) should there be a hat on $f$, and also do you need another Pool given that $\hat{f}$ is already in $NN^{Relu+Pool}$.
5. Theorem 3 iii) should essential support be $[-c,c]^d$ not $[-c,c]$
6. It's not clear why you need the bilinear pooling layer, can you provide more intuition for this?
7. Page 5 Pool equation, should superscript be $n/2$ not $n$, likewise last sentence of following paragraph
8. Why is 'the topology on the latter induced by the L1-norm is strictly stronger than $\tau_{loc}$? This seems important as it shows that $\tau_{loc}$ is not sufficient for your purposes, so should be explained.
9. Where is the fact that $f$ is lipschitz used in Lemma 2 (do you need it)?
10. What is $n$ in Proposition 4?
11. Top line of page 13 doesn't make sense to me, shouldn't it be 'Theorem 1 (ii)' and 'Lemma 6'?
12. I'd suggest rewriting the discussion/adding a conclusion to summarise your contributions rather than delving into higher order derivatives for future work.
13. What do you mean by "parts" in top of page 11.

Typos:
1. "approximate" -> "approximating" in abstract
2. add "than" after "are more popular in practice" in first paragraph.
3. "sharpe" -> "sharp" (also this intuition could be interesting and is not mentioned again, so I would develop what you mean by this).
4. "approximate x" not "approximate a" for paragraph about sec 2.4
5. "prove" not "proof in paragraph below Prop 3.
6. "failure" not "failute" in paragraph below sec 4.
7. "value output of by function" doesn't make sense last paragraph of page 10.
8. $X=[0,1]^d$ just before Lemma 4.

**Strengths And Weaknesses:**

Strengths:
- The paper studies a question, regarding the choice of activation in NNs, that is (to my knowledge) poorly understood and should be of interest to the TMLR community. The paper 'takes a first step towards answering this open question'.
- The paper provides an interesting construction of a novel topology on which PW-Lin NNs are more expressive than analytic NNs, which may be of use in its own right in theoretical studies.
- The paper seems to be technically sound to me.

Weaknesses:
- The paper has multiple complicated definitions and concepts, and while that is inevitable I think the authors can do a much better job to make the paper more readable/accessible to the TMLR community, which will benefit the authors too (see requested changes)
- Correct me if I'm wrong, but it seems the proof to show that analytic NNs are not dense with respect to the new constructed topology boils down to the fact that analytic functions that are compactly supported must be identically zero and that to approximate a function in this new topology we must use functions that have compact support (e.g. proposition 4 and lemma 6). This seems to be a bit simple to me, is that fair to say or have I missed something?
- Intuition for what this separating topolgy $\tau$ (def 2) actually is is somewhat lacking; what is included in $\tau$ that is not present in the standing L1-norm topology $\tau_{\text{norm}}$?
- The construction (provided to answer the question of activation choice) in this work seems to be somewhat far removed from practice though the question itself is of practical relevance; do the authors have any motivation for why a practitioner might find their answer interesting? Note there are results suggesting that classical analytic activations can be trained in practice to as good a better performance as PW-Lin with different initialisation schemes https://arxiv.org/abs/2110.01765, suggesting trainability is one answer rather than expressivity.
- There are quite some typos (see below)

---

> ### Author Response · Authors · 2022-05-29
> **A Streamlined Introduction, Intuition Behind the Separating Topology, and a Tidy Conclusion (Edits in: Pinkish-Purple)**
>
> Dear Reviewer vPRx,
>
> Thank you for your helpful feedback; we feel that your input has helped us make our paper more accessible to the broader TMLR audience.   Below is a list of improvements to our manuscript which we have made following your requested changes:
>
> 1.  Typos and Requested Clarifications "There are quite some typos."
>
>     a. We would like to thank you for taking the time to read our manuscript in detail and to identify several typos or areas passages which could have been better clarified.  We have fixed each of those and other types, and we have clarified points 2-11 and point 13.
>
>    b. Concerning point 1.  We have restructured the paper's introduction and have added an intuitive description of what universal approximation in the separating topology $\tau$ entails, what is most novel about our main results, and what our approximation-theoretic result means/implies.
>
> 2.  "The paper has multiple complicated definitions and concepts, and while that is inevitable, I think the authors can do a much better job to make the paper more readable/accessible to the TMLR community, which will benefit the authors too (see requested changes)."
>
>      a. In the introductory portion of the paper, we have added the subsection "The Qualitative Effect Encoded by the Separating Topology (bottom of page 2 to the middle of page 3).  This section explains the intuition behind our main results, what approximation in the separating topology entails and why deep ReLU networks with bilinear pooling can approximate functions in $L^1_{\operatorname{loc}}(\mathbb{R}^d,\mathbb{R}^D)$ in this sense; while analytic functions cannot.
>
> 3.  "Correct me if I'm wrong, but it seems the proof to show that analytic NNs are not dense with respect to the newly constructed topology boils down to the fact that analytic functions that are compactly supported must be identically zero and that to approximate a function in this new topology we must use functions that have compact support ... is that fair to say or have I missed something?"
>
>    a. We have indeed constructed the separating topology $\tau$ so that compact support is a necessary, but not sufficient, condition for universality in $L^1_{\operatorname{loc}}(\mathbb{R}^d,\mathbb{R}^D)$ for $\tau$.  This is part of the novelty of $\tau$; however, the main contribution of the paper is the interplay between $\tau$ and the model $\operatorname{NN}^{\operatorname{ReLU} + \operatorname{Pool}}$ (this is the main content of Theorem 2, the supporting lemmata, and ultimately Theorem 1 (i) and (iii)).
>
>   b. Indeed, the fact that analytic models are not dense in $L^1_{\operatorname{loc}}(\mathbb{R}^d,\mathbb{R}^D)$ in $\tau$ is an immediate consequence of how we designed $\tau$.  That said, as you also pointed out, its construction is not trivial, and we would like to emphasize that it constitutes a major portion of the mathematical novelty of our paper.
>
> 4.  "Intuition for what this separating topology (def 2) actually is is somewhat lacking; what is included in that is not present in the standing L1-norm topology ?"
>
>    a. We have added a lengthier discussion in the introduction, entitled "The Qualitative Effect Encoded by the Separating Topology," which explains what is captured in the separating topology $\tau$.
>
>   b. A large part of our construction of $\tau$ stems from the topology, which we construct on the space of essentially-compactly supported functions in $L^1(\mathbb{R}^d,\mathbb{R}^D)$ (Section 3, Steps 1-2 on page 10).  This is at the core of our construction, and it relies heavily on the theory of LB-spaces.  We added Example 6, which explains LB-spaces through an example that contrasts an LB-space of polynomial functions against a Banach space of formal power series.
>
> 5.  "The construction (provided to answer the question of activation choice) in this work seems to be somewhat far removed from practice though ... rather than expressivity.  "
>
>      a. This is an entirely fair point that the success of one machine learning model over another, for a specific learning task is due to several factors which constitute its inductive bias; e.g. its approximation-theoretic expressiveness, its ability to generalize well on the specific learning task, and how the training dynamics of the particular optimization algorithm used to train the model to interact with these two factors.  Following your observation, we have included and expanded on this in the discussion section (page 15). We have emphasized that our paper is (to our knowledge) the first to examine the approximation-theoretic side of this problem.
>
> 6 .  "I'd suggest ... adding a conclusion to summarise your contributions".
>
>   a. We have added a conclusion summarizing our contributions, where the novelty lies in our work both from theoretical machine learning and mathematical vantage points.  We also discuss which of the new techniques we introduce in this paper can be used in future research.

---

### Review · Reviewer_jpY7 · 2022-05-12

**Summary Of Contributions:**

The manuscript finds a topology such that with respect to this topology, ReLU neural network class is dense in the locally integrable space while analytic neural network classes are not dense, revealing the advantage of the former to the latter in some sense.
The manuscript highlights the significance of the nonsmoothness of ReLU by showing that polynomial neural networks are also not dense in the locally integrable space with respect to the separation topology. The manuscript presents a quantitative approximation result of ReLU neural network by restricting the target function being in Lipschitz and compactly supported function classes. The result is given in terms of the approximation quality, target’s regularity and the complexity of target’s support.


**Broader Impact Concerns:**

None.

**Requested Changes:**

1.Page 3, line 2, “the essential support...”. Grammar typos.
2.Page 3, theorem 3(i), “$x\in [n_f,n_f]^d$”. Typos.
3.Page 3, theorem 3(iii), the essential support should be contained in a d-dimensional cube. Typos.
4.Page 13, discussion, paragraph 2, ”...the approximation should be in the $C^k$-norm (for some k > 0)”. In PDE settings, we often consider Sobolev spaces instead of continuous function spaces.


At the bottoms of page 2, the authors write “In contrast, both the above
classical methods cannot perform such a simultaneous approximation of a function’s output and its support. ” Can the authors turn this claim into a mathematical proposition with strict proof or cite previous results if it already exists?
The authors mention that the bilinear pooling can be implemented by $ReLU^2$. However, it is worth noting that the bilinear pooling can also be approximated by ReLU as ReLU can approximate multiplication (see [1]).  Does similar results still hold if we remove the pooling layer and add some ReLU layers as compensation?



[1]. Dmitry Yarotsky. Error bounds for approximations with deep relu networks.





**Strengths And Weaknesses:**

Strengths:
 Most of previous literatures on neural network approximation theory consider the approximation in some useful metric spaces, i.e., continuous function spaces, Holder spaces, Sobolev spaces. The manuscript deals with the network approximation issue in a more abstract sense. Specifically, the authors consider approximation in some topology spaces. I believe this is a novel way of studying neural network approximation and I think there are some novel techniques adopted in the proof of the theorems in the manuscript.

Weaknesses :
The authors construct a separation topology but fail to show the optimality of their result. In other words, we don’t know whether there exists a weaker topology which is still able to separate ReLU and analytic neural networks.

---

> ### Author Response · Authors · 2022-05-29
> **Non-Existence of an Optimal Separating Topology and Explanation of the Role of Bilinear Pooling Layer (Edits in Aquamarine)**
>
> Dear Reviewer jpY7,
>
> Thank you very much for your helpful and insightful feedback.  The main changes which you requested are in aqua-marine, in the updated version of our manuscript.
>
> 1. "The authors construct a separation topology but fail to show the optimality of their result.  In other words, we don’t know whether there exists a weaker topology which is still able to separate ReLU and analytic neural networks."
>
> This is a fascinating comment and question.  In our extended discussion section, we added a new proposition (namely Proposition 5) which proves that there does not exist an optimal topology in which ReLU networks with bilinear pooling are dense, but neural networks with analytic activation functions bilinear pooling are dense.  In other words, your comment motivated us to demonstrate that such an optimality result does not exist; we find this conclusion rather nice (and its proof is equally interesting).
>
> 2. "At the bottoms of page 2, the authors write “In contrast, both the above classical methods cannot perform such a simultaneous approximation of a function’s output and its support. ” Can the authors turn this claim into a mathematical proposition with strict proof or cite previous results if it already exists?"
>
> We reorganized the last line of Theorem 2 to emphasize that statement is part of Theorem 2's conclusion; now, it's last line.
>
> 3. The authors mention that the bilinear pooling can be implemented by $ReLU^2$.  However, it is worth noting that ReLU can also approximate the bilinear pooling as ReLU can approximate multiplication (see [1]).  Do similar results still hold if we remove the pooling layer and add some ReLU layers as compensation?
>
> We added to the discussion section, specifically at the bottom of page 15 and the top of page 16, to address this question.  Namely, we emphasize that even if Yarotsky or Kidger and Lyons construction constructs a ReLU network which can approximately implement multiplication, our construction really needs exact implementation of this operation.  We explain why this is the case by looking in more detail at the construction of the approximating neural network in Theorem 2.  The construction builds a "masking ReLU network" which has values 0 outside $\cup_{i=1}^{n+1}\, K_i$, value $1$ in $\cup_{i=1}^{n}\, K_i$, and intermediate values in between, an "approximating ReLU network" which uniformly approximates the target function $f$ on the compact set $\operatorname{ess-supp}(f)$ and it uses the bi-linear pooling layers to multiply these two networks to obtain the desired effect.  We emphasize that the subtlety here is that if this multiplication is only approximate (on compact sets); thus without bilinear pooling (and only using the now classical approximate multiplication construction which you mention) there is no guarantee that the resulting network has appropriately controlled compact support; whence it may fail to approximate $f$ in the separating topology $\tau$.
>
> We have also corrected all the typos which you have pointed out.

---

> > ### Comment · Reviewer_jpY7 · 2022-06-24
> > **Sorry for delay.**
> >
> > Thanks the reviewer for their response. I'm recommending for accept.

---

### Review · Reviewer_7i2i · 2022-05-18

**Summary Of Contributions:**

The paper proposes a topological approach to understand the differences between neural networks with piecewise linear vs. analytic activations. The results are collected in three theorems. Theorem 1 states the existence of a topology on the space of locally integrable functions with respect to which networks with piecewise linear activations are dense while those with analytic activations are not. Theorem 2 states that polynomials are not dense in this topology, either. Theorem 3 is a quantitative version of Theorem 1 providing specific bounds on the approximation by a piecewise linear network in terms of its support, accuracy and network size.

**Broader Impact Concerns:**

No concerns

**Requested Changes:**

1. Explain why Theorem 1 is nontrivial and the two types of networks cannot be swapped in it.

2. Give a more accessible explanation of the proposed topology (e.g., if possible, describe its open sets in terms of usual metric open sets) and associated convergence. Explain why the main result is more than just the observation that analytic functions cannot be compactly supported. Give at least a heuristic explanation for Proposition 1 since it seems to play a significant role in the results.

3. Fix the issue with the density described in terms of convergence of sequences.

**Strengths And Weaknesses:**

I think that the overall idea of the proposed topological approach, as well as some specific technical contributions (the separating topology, "good partitions" and Proposition 1, new complexity bounds involving target's support) are interesting. However, I see some serious issues with the paper in the present state.

1. I don't understand why Theorem 1 is nontrivial as formulated. Suppose that we have a dense set A in a topological space, and another set B that does not contain A. Then, as far as I understand, it is always possible to construct a refined topology in which A is dense and B is not: simply add the single-point sets for all elements of A to the topology base. Moreover, when applied to the present setting of neural networks, this works both ways, i.e. it is also possible to construct a topology in which networks with analytic activations are dense while those with piecewise linear activations are not. This point is not discusssed in the paper at all, but it conflicts with its general sentiment that one kind of activations has a fundamental advantage over another.

2. The paper provides a particular construction for its topology, based on limit-Banach spaces. This construction is not easy to digest. Some of its essential properties even require category theory (Proposition 1). At the same time, the practical meaning of this topology and its relevance to approximation by neural networks are not intuitively clear. It appears from Theorem 3 that the crux of the matter is that this topology requires approximating functions to have a compact support. However, it is obvious that networks with analytic activations cannot implement nontrivial functions with compact support. This is a very simple idea already showing the significant difference between piecewise linear and analytic activations, and the added value of the whole presented complicated topological framework is not very clear to me.

3. It is claimed in the end of section 2.3 (and later used) that the density of a subset in a topological space is equivalent to all points of the space being limits of sequences from the subset. This is not true in general (for non-metrizable topological spaces); one has to consider nets instead of sequences.

---

> ### Author Response · Authors · 2022-05-29
> **Improvement of Theorem 1, Explanation of the Qualitative Effect Captured by the Separating Topology, and Emphasis on Main Contributions vs Simple Consequences (Edits in: Blueish-Silver)**
>
> Dear Reviewer 7i2i,
>
> Thank you for your insightful and helpful feedback.  We believe that the changes you requested, highlighted in blueish-silver in the updated version of the manuscript, have strengthened and clarified the main contributions of our paper.  These changes are as follows:
>
> 1.  “Explain why Theorem 1 is nontrivial, and the two types of networks cannot be swapped in it.”
>
>       a. We have added a point (iii) in Theorem 1, which states that the separating topology $\tau$ does not contain any open singleton sets of the form {f}, where $f$ is a piecewise linear neural network with bi-linear pooling layers.  The proof of Theorem 1 (iii) has been added to the end of the appendix.
>
>      b.  We have explained the implication of Theorem 1 (iii) following Theorem 1.  Namely, it excludes requiring forcing all “universal models” to be able to implement some piecewise linear neural networks with bi-linear pooling layers.  Mathematically speaking, it prohibits obtaining Theorem 1 (i) and (ii) by simply adding some non-empty set of piecewise linear neural networks with bi-linear pooling layers to the basis of $L^1_{\operatorname{loc}}(\mathbb{R}^d,\mathbb{R}^D)$; thus, as you rightly pointed out, it shows that both network architectures’ roles in Theorem 1 cannot be swapped and the theorem remains valid.
>
> 2.  “Give a more accessible explanation of the proposed topology” (note this is coloured in pink-purple; since it is also a question of reviewer vPRx)
>
>       a. We added a description of the separating topology in the introduction section at the end of page $2$ and at the beginning of page $3$; this also included moving Figure 1 up from inside the text’s main body here; where we use it to illustrate the effect of function approximation in $\tau$.
>
>      b. In Section 2.4 (page 9), we have added an extended example that we feel illustrates the LB-space construction, which is central to the construction of the separating topology $\tau$.  Namely, the example shows how the set of all real-valued polynomial functions on $\mathbb{R}$ can be made into an LB-space by gluing together Euclidean spaces, each of which contains all polynomials of up to a given degree.  The example then contrasts this against the smallest Banach space containing all polynomial functions and each Euclidean space of polynomials up to a given degree as subspaces.  We show that the latter contains many power series while the former does not, which illustrates the “smallness” of our construction.
>
>     We also then given an explicit characterization of any open convex set in that LB-space; as a means to illustrate what open convex sets must look like in any LB-space.
>
>     c. Along with Theorem 1 (iii), we emphasized in our modified discourse to emphasize that the key theoretical innovations here are the construction of a family of ReLU networks with bi-linear pooling layers, which are dense in the topology $\tau$ and the qualitative effect described by $\tau$.  We also now emphasize in our writing (e.g. on page $3$ in pink-purple) that $\tau$ is built such that it is necessary but not sufficient for a universal model to contain an expressive subset of compactly supported models; thus, the non-universality of analytic models is a simple consequence of our main results.  Namely, the construction of $\tau$ and the universality of ReLU networks with bi-linear pooling layers in $\tau$.
>
>     d. We have added a heuristic explanation of Proposition 1 circa equation (4).  We discuss why the proposition is helpful, what it expresses, and why the result is intuitive (even if it can be mathematically challenging to show, as you noted, the use of Category Theory).
>
>    e.  The proof of Theorem 1 (iii) also uses an explicit description of open sets in the separating topology $\tau$.  However, we preferred not to include this in the manuscript's main body; as the description is more technical than it is insightful; we hope you also feel the same way.
>
> 3.  ``Fix the issue with the density described in terms of convergence of sequences.''
>
>     We have fixed the issue with the convergence of sequences in the background section.  Thank you very much for pointing this out. We feel that we had oversimplified this part of the discussion to the point of it no longer being correct in the previous iteration.

---

> > ### Comment · Reviewer_7i2i · 2022-06-14
> > **Comments on the revised manuscript**
> >
> > I thank the authors for their response. The revised version contains a number of improvements, but still has many issues and is not ready for publication.
> >
> > 1.
> > * I disagree that the updated version of Theorem 1 is nontrivial and that the two models cannot be swapped in it. In my initial review I referred to a construction involving single-point open sets, but it's just one way to refine the topology; there are other possibilities not involving single-point sets. To be specific, consider the refined topology of $L^1_{\mathrm{loc}}(\mathbb R^d,\mathbb R^D)$ having open sets of the form $O_1 \cup (O_2\cap (\mathrm{NN^{\omega+Pool}}\setminus\mathrm{NN^{\sigma_{PW-lin}+Pool}}))$, where $O_1, O_2$ are open in the original topology. This topology fulfills all statements of Theorem 1 (including new statement iii) with the roles of the models $\mathrm{NN^{\omega+Pool}}$ and $\mathrm{NN^{\sigma_{PW-lin}+Pool}}$ reversed, i.e. $\mathrm{NN^{\omega+Pool}}$ is now dense while $\mathrm{NN^{\sigma_{PW-lin}+Pool}}$ is not dense.
> >
> > * (Most important) I don't think that the authors have properly addressed the key issue that I indicated, namely that their main message regarding the fundamental gap between the two models is essentially wrong. This message is emphasized  repeatedly in the manuscript, e.g., citing from the conclusion: "This paper showed that deep feedforward networks with piecewise linear activation functions and utilizing bilinear pooling layers are strictly more expressive than deep feedforward networks deploying any number of analytic activation functions and leveraging bilinear pooling layers." You cannot claim that one model is strictly more expressive than another if the main result supposedly showing this remains valid with the roles of the two models reversed, i.e. the relation between the models is essentially symmetric. I do not understand why this point is not discussed in the paper. The added statement (iii) in Theorem 1 does not address this symmetry. The manuscript has technical merit in describing a particular topology favoring $\mathrm{NN^{\sigma_{PW-lin}+Pool}}$, but is misleading in its broad claims.
> >
> > 2. A key point in the sketch of proof added for Proposition 1 is based on compactness of $\cup_{i=1}^{n+1}K_i$, but there are no assumptions of compactness of $K_i$ in the definition of good a.e. partition (Definition 1).
> >
> > 3. There is again a wrong statement in the end of section 2.3: "if $A\subset X$ is such that there is some $x\in X$ such that there is not sequence $(x_n)_{n=1}^\infty$ of elements in $A$ which converges to $x$ in $\tau_X$, then $A$ is not dense in $X$ with respect to $\tau_X$". A standard counterexample is the cocountable topology: let $X$ be uncountable and the nonempty open sets be the complements of the countable subsets of $X$. Then any proper uncountable subset $A$ is dense in $X$, but for any $x\in X\setminus A$ there is no sequence in $A$ converging to $x$.

---

> > > ### Author Response · Authors · 2022-06-15
> > > **Incorporation of Proposition 4 and Lemma 3's statements into Theorem 1 $\rightarrow$ Non-Interchangeability of Models in Theorem 1 + Added Description of the separating topology $\tau$**
> > >
> > > Dear Reviewer 7i2i,
> > >
> > > 1) We have addressed your point 1, by incorporating the statements of Lemma 3 and Proposition 4 directly into Theorem 1.  This ensures that the roles of the two models cannot be swapped; thus addressing your concern.
> > > Furthermore, it has the added advantage of offering some insight into what the separating topology $\tau$ is describing.
> > >
> > > 2) As you pointed out, we used but didn’t state the compactness of the sets forming the good a.e. partition of $\mathbb{R}^d$.  We have corrected this typo in Definition 1.  Thank you.
> > >
> > > 3) We have added the complete description of convergence in general topological spaces, using nets.  We admit that the lack of rigour here was induced by attempting to oversimplify a technical concept.
> > >
> > >
> > > We feel that our modifications (in Teal) have addressed your main concerns.  Please let us know if you feel your concerns have been addressed with these modifications.
> > >
> > > Best,
> > > The Authors

---

> > > > ### Author Response · Authors · 2022-06-19
> > > > **Calibration of Writing to Theoretical Results: ReLU Networks are Well-Suited to Approximation of Compactly-Supported Locally-Integrable Functions but Need-Not for Other Learning Tasks**
> > > >
> > > > Dear Reviewer 7i2i,
> > > >
> > > > We believe that much of the contention over our Theorem 1 is due to the miscalibration of the writing in our manuscript at some points with the results which we derive.
> > > >
> > > > To explain our changes let us briefly discuss what we have shown in our paper.
> > > >
> > > > *Summary of What We Have shown*
> > > >
> > > > We showed that, unlike general locally Lebesgue-integrable functions, essentially compactly-supported Lebesgue locally-integrable functions carry additional structure that one may want to represent by a deep learning model.  Thus, one may hope that a deep learning model can not only approximate such functions in local-$L^1$ or $L^1$ (as with general locally integrable functions) but that one can do so while implementing these function’s essential support; at least up to a discretization of the input space (formalized by our good a.e. partitions on $\mathbb{R}^d$).
> > > >
> > > > Accordingly: Theorem 1 shows
> > > >
> > > > - Points (iii) and (iv) There is a topology $\tau$ on $L^1_{\operatorname{loc}}(\mathbb{R}^d,\mathbb{R}^D)$ which formalizes this notion of approximation; which captures much of the additional structure of essentially-compactly supported functions,
> > > >  - This notion of universal approximation extends to the classical notion when considering general (not compactly supported) Lebesgue locally-integrable functions
> > > > - Point (i) ReLU with bilinear pooling networks are universal in this sense *(thus are well-suited to the approximation of essentially compactly-supported Lebesgue locally-integrable functions)*,
> > > > - Point (ii) Networks with analytic activation functions (even with pooling) may be ill-suited to the approximation of compactly supported functions since they fail to approximate their value and their support simultaneously.
> > > >
> > > > Theorem 2 then makes Theorem 1 quantitative on the dense and “more regular” subclass of compactly supported Lipschitz functions.
> > > >
> > > >
> > > > *Summary of Our Writing Edits*
> > > >
> > > > - We have calibrated the writing to emphasize the novelty of this claim and not to claim that "ReLU networks are fundamentally better than networks with analytic activations functions".
> > > > - We have emphasized the novelty of our quantitative universal approximation theorem (Theorem 2) which can incorporate the metric capacity of the support of a compactly-supported Lipschitz functions into its complexity estimates (depth, width, and the number of bilinear pooling layers), and e.g. our other constructions such as good a.e.
> > > > - We have added a subsection 5.1 to the discussion section 5, which clarifies that we are not claiming that ReLU networks are always better but rather that they are more suitable for the approximation of compactly-supported Lebesgue locally-integrable functions; if one holds that the approximation of these functions should at-least imply their $L^1$-approximation and the implementation of their support; at-least up to a discretization of the input space.
> > > > - We have placed more emphasis on subsection 5.2 which describes a class of problems in which ReLU networks must be ill-suited while networks with analytic activation functions need not be (i.e. physics-informed learning when one needs (weak) higher partial derivatives).
> > > >
> > > > We believe that these writing modifications, in addition to the technical improvements to our manuscript from the previous comment, address our concerns and have also improved the overall quality of our submission.
> > > >
> > > > Thank you very much for taking the time to review our manuscript and for the helpful discussion on OpenReview.
> > > >
> > > > Best regards,
> > > > The Authors

---

### Decision · Action_Editors · 2022-08-02

**Recommendation:** Accept as is

**Comment:**

I would like to thank the reviewers for a first round of on-time reviews. I would like to extend gratitude to one particular reviewer for their diligence in identifying aspects of the original presentation which would have likely obscured the nature of the contributions to the community.

After distilling some of the changes that I saw would be necessary to address this reviewer's concerns, the authors agreed on a major revision. This was carried out with such speed that the key reviewer who raised the original issues around presentation was able to re-review this updated version without the need for a formal second round. The reviewer found the changes acceptable and pointed out a few minor issues. The authors have addressed these minor issues. I also agree that the major revision addressed the key issues and find the article acceptable in its present form.

---

> ### Author Response · Authors · 2022-08-03
> **Thanks to all the reviewers and to the action editor for their work**
>
> Dear Reviewers and action editor,
>
> We would like to thank you all very much for your very helpful feedback and for this very pleasant review process. We feel that the paper benefited greatly from the entire process and are grateful for all your hard work.
>
> Best regards,
>
> Anastasis and Behnoosh